# Cohesion is established during DNA replication utilising chromosome associated cohesin rings as well as those loaded de novo onto nascent DNAs

Madhusudhan Srinivasan[1]*, Marco Fumasoni[2], Naomi J Petela[1], Andrew Murray[2], Kim A Nasmyth[1]*

[1]Department of Biochemistry, University of Oxford, Oxford, United Kingdom; [2]Department of Molecular and Cellular Biology, Harvard University, Cambridge, United States

**Abstract** Sister chromatid cohesion essential for mitotic chromosome segregation is thought to involve the co-entrapment of sister DNAs within cohesin rings. Although cohesin can load onto chromosomes throughout the cell cycle, it only builds cohesion during S phase. A key question is whether cohesion is generated by conversion of cohesin complexes associated with un-replicated DNAs ahead of replication forks into cohesive structures behind them, or from nucleoplasmic cohesin that is loaded de novo onto nascent DNAs associated with forks, a process that would be dependent on cohesin's Scc2 subunit. We show here that in *S. cerevisiae*, both mechanisms exist and that each requires a different set of replisome-associated proteins. Cohesion produced by cohesin conversion requires Tof1/Csm3, Ctf4 and Chl1 but not Scc2 while that created by Scc2-dependent de novo loading at replication forks requires the Ctf18-RFC complex. The association of specific replisome proteins with different types of cohesion establishment opens the way to a mechanistic understanding of an aspect of DNA replication unique to eukaryotic cells.

*For correspondence:
madhusudhan.srinivasan@bioch. ox.ac.uk (MS);
ashley.nasmyth@bioch.ox.ac.uk (KAN)

**Competing interests:** The authors declare that no competing interests exist.

## Introduction

The error-free duplication and segregation of chromosomal DNAs is fundamental to cell proliferation. Sister DNAs generated during DNA replication must be held together from S phase until anaphase, when they finally disjoin to opposite poles of the cell. While there are similarities in the way bacteria and eukaryotes replicate their genomes, the establishment of sister chromatid cohesion during S phase and its destruction during anaphase are aspects of chromosome biology unique to eukaryotic cells. Cohesion is mediated by cohesin, which belongs to one of three classes of eukaryotic SMC-kleisin protein containing complexes (*Yatskevich et al., 2019*).

Although initially identified as being necessary for sister chromatid cohesion (*Guacci et al., 1997*; *Michaelis et al., 1997*), cohesin also has a major role in organizing the topology of individual interphase chromatin fibres (*Rao et al., 2017*), a function thought to stem from its ability to trap and processively enlarge DNA loops (*Davidson et al., 2019*; *Kim et al., 2019*). The notion that most cohesin associated with un-replicated DNA is engaged in extruding loops raises an important conundrum, namely what alteration in its enzymatic activity is required to establish cohesion between nascent sister DNAs during S phase. Elucidating this problem will require a better understanding of the molecular mechanisms involved in both processes. Like all other SMC protein complexes, cohesin contains a pair of rod-shaped SMC proteins (Smc1 and 3), whose association via their hinge domains creates V-shaped heterodimers with ATPase domains at their vertices that are interconnected by a kleisin subunit (Scc1) to form trimeric rings. These trimers have little or no ATPase or

loop extrusion activity in vitro nor do they possess any biological function in vivo and require an additional set of proteins, namely a set of hook-shaped proteins composed of HEAT repeats known as HAWKs (HEAT repeat proteins Associated With Kleisins) (*Wells et al., 2017*). Cohesin has three HAWKs: Scc3 is permanently bound while Scc2/Nipbl and Pds5 appear interchangeable (*Petela et al., 2018*).

Along with the finding that cleavage of Scc1 by separase triggers cohesin's dissociation from chromosomes as well as sister chromatid disjunction at the onset of anaphase (*Oliveira et al., 2010*; *Uhlmann et al., 1999*), the discovery that cohesin forms a tripartite ring raised the possibility that it holds DNAs together using a topological principle, by co-entrapment of sister DNAs within Smc-kleisin rings, a notion known as the ring model (*Haering et al., 2002*; *Nasmyth, 2001*). Testing this has been made possible through elucidation of the molecular structures of the Smc1/Smc3, Smc3/Scc1, and Scc1/Smc1 interfaces, which has permitted the insertion of cysteine pairs that can be chemically crosslinked in vivo as well as in vitro using bi-functional thiol-specific reagents. As predicted by the ring model, crosslinking all three interfaces in living cells creates circular Smc1-Smc3-Scc1 polypeptides within which circular mini-chromosome DNAs would be entrapped in a manner resistant to SDS treatment. Catenation by cohesin of individual DNAs in this manner modestly retards their electrophoretic migration (creating catenated monomers or CMs) while co-entrapment of sister DNAs causes them to migrate as dimers (catenated dimers or CDs) even when they are not otherwise intertwined through DNA-DNA catenation (*Gligoris et al., 2014*; *Haering et al., 2008*; *Srinivasan et al., 2018*).

Strict topological entrapment of DNAs within cohesin rings, as measured in the above manner, is not essential either for cohesin's association with yeast chromosomes in vivo (*Srinivasan et al., 2018*) or for loop extrusion by mammalian cohesin rings in vitro (*Davidson et al., 2019*). Nevertheless, cohesin is capable of entrapping individual DNAs prior to DNA replication and also does so when expressed in post-replicative cells, where it is known to load onto chromosomes without forming sister chromatid cohesion (*Srinivasan et al., 2019*; *Srinivasan et al., 2018*). In contrast, CD formation accompanies the establishment of cohesion in all cases hitherto analysed, suggesting that co-entrapment within cohesin's Smc-kleisin ring may indeed be the mechanism by which cohesin holds sister DNAs together.

Throughout most of the cell cycle, apart from anaphase when kleisin cleavage causes rapid dissociation, the fraction of cohesin associated with chromatin is determined by its rate of association and its rate of dissociation. The former depends on Scc2, Scc3, and Scc4, which binds to Scc2's N-terminal domain (*Chao et al., 2015*; *Hinshaw et al., 2015*; *Kikuchi et al., 2016*), while the latter depends on a regulatory subunit called Wapl, which binds to both Scc3 and Pds5 (*Yatskevich et al., 2019*).

Scc2 is not merely a loading factor as it is also necessary for maintaining loop extrusion in vitro (*Davidson et al., 2019*) as well as cohesin's association with chromatin in vivo during G1 phase (*Srinivasan et al., 2019*). In the latter case, it does so by preventing dissociation of the Smc3/Scc1 interface (*Srinivasan et al., 2019*). How Wapl overrides this inhibition and thereby triggers release is not understood. What is clear is that the maintenance of sister chromatid cohesion, once established during S phase, depends on abrogation of Wapl-induced release through modification of K112 and K113 on Smc3 by the Eco1 acetyl transferase (*Rolef Ben-Shahar et al., 2008*; *Ivanov et al., 2002*; *Unal et al., 2008*). In yeast, acetylation only takes place during S phase and appears to be sufficient to block release (*Beckouët et al., 2010*). In mammalian cells, Smc3 acetylation, though necessary, is insufficient and the continual association of another factor named sororin is necessary to protect cohesive structures over the long term (*Nishiyama et al., 2010*).

Because it is required to build cohesion during S phase but not to maintain it during G2 or indeed for cohesin's association with chromatin in the first place, Eco1 was initially thought to be intimately involved in the process by which cohesion is established (*Toth et al., 1999*). It is now appreciated that these acetyl transferases are merely needed to counteract Wapl-mediated release and as a consequence they are largely dispensable for building and maintaining sister chromatid cohesion in yeast (*Rolef Ben-Shahar et al., 2008*; *Chan et al., 2012*; *De et al., 2014*; *Feytout et al., 2011*; *Rowland et al., 2009*; *Srinivasan et al., 2018*; *Sutani et al., 2009*), mammalian (*Ladurner et al., 2016*) or even plant cells (*De et al., 2014*) lacking Wapl.

Although it is not known whether cohesive structures throughout the genome involve co-entrapment of sister DNAs within cohesin's Smc-kleisin ring as observed with mini-chromosomes (i.e. CD formation), this possibility is entirely plausible. Even if not, co-entrapment may be created by a

similar if not identical process to actual cohesion and is therefore an excellent surrogate (*Srinivasan et al., 2018*). Crucially, it is currently the only simple physical measure of cohesion. How these structures are created remains poorly understood. An important finding is that cohesin which loads onto chromosomes in post-replicative cells does not tether sister DNAs together, even though it is capable of entrapping individual DNAs (*Haering et al., 2004*; *Srinivasan et al., 2019*). The process of establishing cohesion appears therefore to be confined to S phase. A key question is whether cohesive structures or CDs are derived from complexes previously associated with un-replicated DNA or created de novo by ones that load during the passage of replication forks. We will henceforth refer to these two scenarios as conversion and de novo loading, respectively (*Figure 1A*).

The first indication that chromosomal cohesin associated with un-replicated DNA can be 'converted' into cohesive structures was the observation that replication fork passage in mammalian cells does not cause dissociation of chromosomal cohesin, at least in the absence of Wapl (*Rhodes et al., 2017*). In this case, however, chromosomal cohesin was merely observed optically and it was not established whether it was turned into cohesive structures during the passage from G1 to G2. More recently, we have shown that separase resistant cohesin complexes loaded onto yeast mini-chromosomes during G2 remain associated with chromatin and are converted into cohesive structures upon a round of re-replication induced by the transient inhibition of Cdk1. (*Srinivasan et al., 2019*). Crucially, this occurs even when Scc2 has been inactivated after the initial loading during G2, suggesting that cohesin is converted into cohesive structures without any further participation of Scc2 (*Srinivasan et al., 2019*). However, these experiments did not exclude the possibility that cohesion

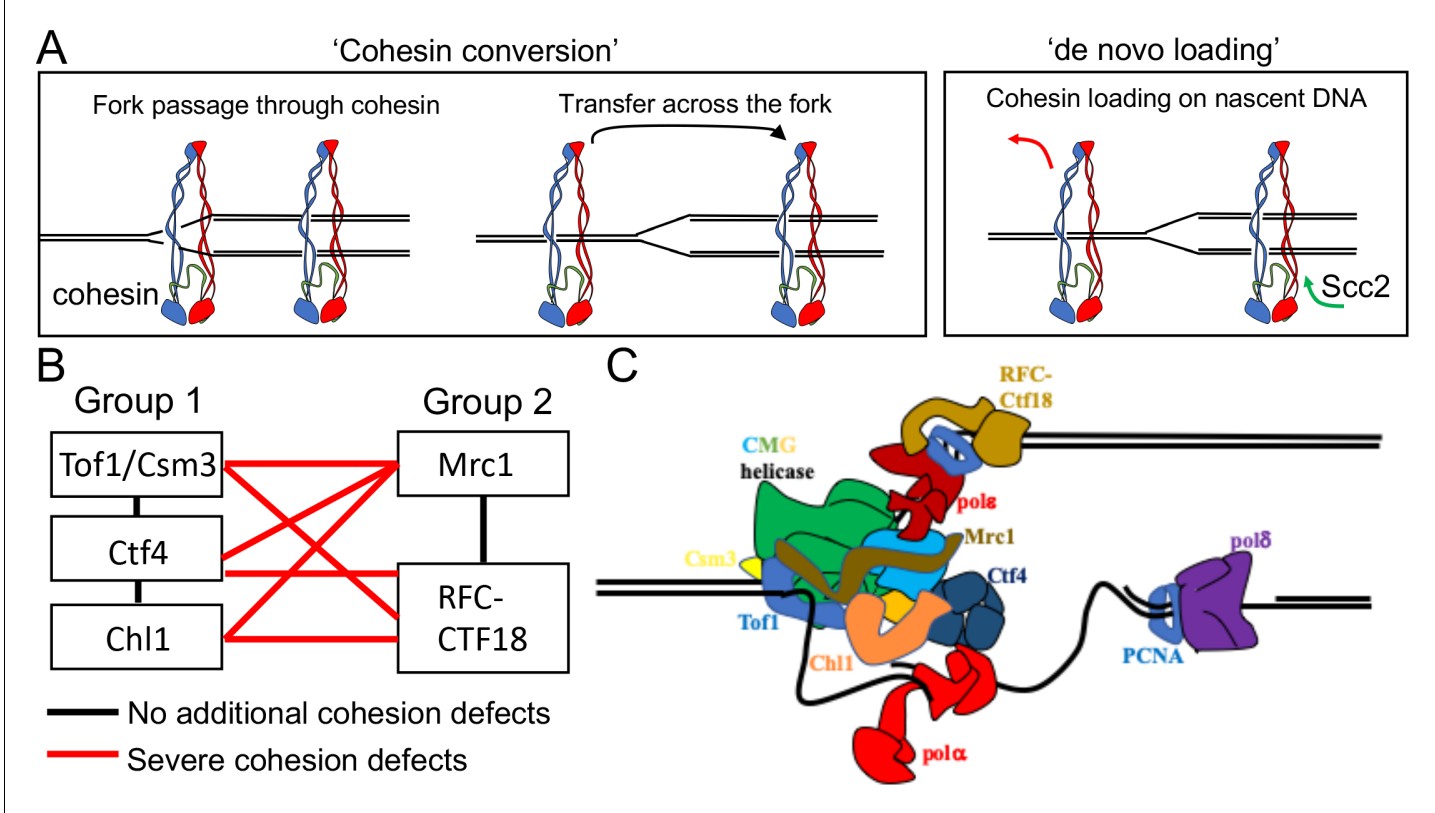

**Figure 1.** Possible mechanisms of cohesion establishment. (**A**) Cohesin complexes previously associated with un-replicated DNA converted into cohesive ones behind the replication fork (Cohesin conversion) or cohesion built by de novo loading of cohesin molecules onto nascent DNAs at the replication forks (de novo loading).(**B**) The replisome-associated proteins that affect cohesion establishment belong to two epistasis groups, one containing Chl1, Ctf4, Csm3 and Tof1 and a second containing Mrc1 as well as the Ctf18-RFC. Combining two mutants from the same group has little or no additional effect on viability and causes no greater cohesion defect than the single mutants, while combining mutants from different groups either causes lethality or synthetic sickness and, when measured using conditional alleles, greatly increased defects in sister chromatid cohesion. Data from *Xu et al., 2007*. (**C**) The approximate positions of the Tof1/Csm3, Ctf4, Chl1, Mrc1 and the Ctf18-RFC within the replisome is shown (*Baretić et al., 2020*; *Grabarczyk et al., 2018*; *Simon et al., 2014*; *Villa et al., 2016*).

might also be generated through Scc2-dependent de novo loading during S phase, a possibility raised by in vitro studies (*Murayama et al., 2018*).

The notion that cohesion might be established during S phase by two different pathways is consistent with the finding that numerous proteins associated with replisomes, namely Ctf4 (*Hanna et al., 2001*), Chl1 (*Petronczki et al., 2004*; *Skibbens, 2004*), the Csm3/Tof1 complex (*Mayer et al., 2004*), Mrc1 (*Xu et al., 2004*), and the Ctf18-RFC *Mayer et al., 2001* have important albeit non-essential roles in cohesion establishment. A comprehensive analysis of the viability of double mutant strains as well as their sister chromatid cohesion implied that these genes/proteins belong to two epistasis groups, one containing Chl1, Ctf4, Csm3 and Tof1 and a second containing Mrc1 as well as Ctf8, Ctf18, and Dcc1, which are all subunits of the Ctf18-RFC (*Xu et al., 2007*). Thus, combining two mutants from the first group or from the second group has little or no additional effect on viability and causes no greater cohesion defect than the single mutants, while combining mutants from different groups either causes lethality or synthetic sickness and, when measured using conditional alleles, greatly increased defects in sister chromatid cohesion (*Figure 1B*). Although none of these proteins are necessary for DNA replication, all are associated with the replisome, suggesting that their role is to establish cohesion during S phase. Accordingly, inactivation of conditional alleles during G2/M has no effect on the maintenance of cohesion once established (*Xu et al., 2007*). Moreover, expression of Chl1 during G2/M cannot rescue the cohesion defects caused by replication in its absence (*Petronczki et al., 2004*). Though not essential, all of these replisome proteins are highly conserved among eukaryotes and aid cohesion establishment in human as well as yeast cells (*Cortone et al., 2018*; *Zheng et al., 2018*). Structural studies have located with great precision the position of several of these proteins within replisome (*Figure 1C*), but how they contribute to cohesion establishment is currently unclear.

We show here that Chl1, Ctf4, and Csm3/Tof1 are essential for converting cohesin associated with un-replicated DNA into cohesive structures holding together sister DNAs, whereas Mrc1 and Ctf18-RFC are not. Crucially, cohesin conversion mediated by Chl1, Ctf4, and Csm3/Tof1 is independent of Scc2. We present evidence that Ctf18-RFC is instead required for cohesion established through de novo loading at forks and show that this Scc2-dependent process (but not cohesin conversion) is enhanced by modest over-expression of Scc2/4. Our data are fully congruent with the previous suggestion that Mrc1 and Ctf18-RFC are involved in a separate pathway to that of Chl1, Ctf4, and Csm3/Tof1 (*Xu et al., 2007*). Moreover, the discovery of two physiologically as well as genetically distinct cohesion establishment pathways explains why neither pathway is essential and finally opens the way toward a mechanistic understanding of an aspect of S phase that is unique to eukaryotic cells.

## Results

### Cohesin associated with un-replicated chromosomes is converted to cohesive structures in the absence of Scc2 activity

Addressing whether cohesin rings that entrap sister DNAs after replication are derived from those associated with un-replicated chromosomes is complicated by the fact that chromosomal cohesin is turned over rapidly during G1 by Wapl and for this reason we conducted all experiments in cells in which Wapl-dependent release has been abrogated. To exclude any possibility of cohesion being generated from cohesin loaded during the process of DNA replication, we needed to test its dependence on Scc2, which is essential for de novo loading. However, doing this with endogenous wild type cohesin is not in fact possible because Scc2 is also necessary for maintaining cohesin's stable DNA association during G1, even in cells lacking Wapl-dependent release (*Srinivasan et al., 2019*). Thus, inactivation of Scc2 during G1 causes cohesin to dissociate from chromosomes, which is incompatible with observing its conversion into cohesive structures. Two previous observations have been crucial to overcoming this problem. First, cohesin expressed in post-replicative G2 cells can load onto chromosomes and entrap individual DNAs (CMs) but cannot form cohesion or entrap sister DNAs (CDs) (*Haering et al., 2004*; *Srinivasan et al., 2019*). Second, unlike cohesin loaded during G1, cohesin loaded during G2/M does not require Scc2 to prevent its release from chromosomes and remarkably this property is maintained when cells exit mitosis and enter a new G1

phase, as long as the kleisin subunits of the cohesin in question are refractory to separase-mediated cleavage, which would otherwise induce its destruction (*Srinivasan et al., 2019*).

These findings enabled us to measure the establishment of cohesion generated exclusively by cohesin previously associated with un-replicated chromosomes (*Figure 2A*). Our assay relies on the covalent circularization using the homo-bifunctional crosslinker BMOE of '6C' cohesin complexes that contain cysteine pairs at all three of the ring's interfaces (2CScc1 2CSmc1 2CSmc3) and entrapment within these rings of individual or co-entrapment of circular sister mini-chromosome DNAs, producing catenated monomers (CMs) or dimers (CDs) respectively, whose electrophoretic migration following SDS treatment at 65°C is modestly (CM) or greatly (CD) retarded compared to naked mini-chromosome DNA (*Srinivasan et al., 2019*; *Srinivasan et al., 2018*).

A crucial feature of the assay is that endogenous *SMC1* and *SMC3* genes are replaced by fully functional versions that express 2CSmc1 and 2CSmc3 while non-cleavable PK-tagged 2CScc1$^{NC}$ (*Uhlmann et al., 1999*) is expressed ectopically from the *GAL1-10* promoter in cells whose propagation is sustained by a wild-type *SCC1* gene. Thus, only 2CScc1$^{NC}$ expressed from the *GAL* promoter can form 6C cohesin capable of producing CMs or CDs. It is also important to point out that the measurement of CMs and CDs using gel electrophoresis is performed on DNAs precipitated using PK-specific antibodies. Thus, DNAs migrating as supercoiled monomers were also bound by 2CScc1$^{NC}$ but had failed to be entrapped in a fashion resistant to SDS, either because they were not entrapped or because covalent circularization of 6C cohesin is incomplete (only 20–25% are crosslinked at all three interfaces).

Wild type (*SCC2*) and *scc2-45* (temperature sensitive) mutant cells were first arrested in G2/M by treatment with the spindle poison nocodazole at 25°C. 2CScc1$^{NC}$ was induced by galactose transiently (for 45 min) and further expression subsequently repressed (by replacing galactose by glucose) for the remaining course of the experiment. The level of CMs produced in the G2/M *SCC2* and *scc2-45* cells was very similar as was the level of naked supercoiled DNAs associated with 2CScc1$^{NC}$ (but not covalently entrapped) (*Figure 2B*) and, as expected, no CDs were produced. Both cultures were then allowed to complete mitosis, by removing the nocodazole, and subsequently blocked in G1 by incubation in the presence of the mating pheromone α-factor. Because it is not cleaved by separase, the 2CScc1$^{NC}$ cohesin loaded onto the mini-chromosomes during G2/M remained associated with them during this transition, an appreciable fraction of it as CMs (*Figure 2B*). Moreover, calibrated ChIP sequencing showed that although there were variations in cohesin occupancy along the chromosomes, presumably due to cohesin re-localization, the median level of association along the entire genome of this pool of cohesin did not alter while cells remained blocked for an extended period in G1 (*Figure 2—figure supplement 1A and B*). Because Wapl-mediated release is abrogated in these cells and because there is no further synthesis from the *GAL* promoter, the constant level of chromosomal 2CScc1$^{NC}$ cohesin during the arrest implies that the G1 cells inherit little or no free nucleoplasmic 2CScc1$^{NC}$ cohesin that could load onto DNA during this period.

To measure the fate of this 2CScc1$^{NC}$ cohesin pool during DNA replication in the presence (*SCC2*) or absence (*scc2-45*) of Scc2 activity, both cultures were transferred to pheromone-free medium at 37°C, whereupon both sets of cells duplicated their genomes and produced CDs in similar amounts (*Figure 2B*). In the *scc2-45* cells, this was accompanied by a reduction in the fraction of DNAs that formed CMs, raising the possibility that the CMs associated with un-replicated DNAs were converted to CDs upon replication. Higher levels of CMs were observed in the *SCC2* wild-type cells, suggesting that some de novo loading may occur, either during S phase or during the subsequent G2. These data imply that cohesin associated with un-replicated DNAs is converted to cohesive structures (as measured by CD formation) in a fully Scc2-independent fashion. Because not all un-replicated mini-chromosome DNAs associated with 2CScc1$^{NC}$ are present as CMs, we cannot exclude the possibility that some the CDs are produced by chromosomal cohesin rings that had not in fact previously entrapped DNAs. We conclude that cohesin associated with un-replicated mini-chromosome DNAs is converted into cohesive structures during S phase in the absence of Scc2.

## Chl1, Ctf4, and Csm3/Tof1 are essential to establish cohesion from cohesin associated with un-replicated chromosomes

We next addressed if any of the non-essential replisome proteins previously implicated in cohesion establishment, namely Mrc1, Chl1, Ctf4, Csm3/Tof1 and Ctf18, are required for cohesin conversion. Before doing so, we first measured whether loss of these proteins has any appreciable effect on CD

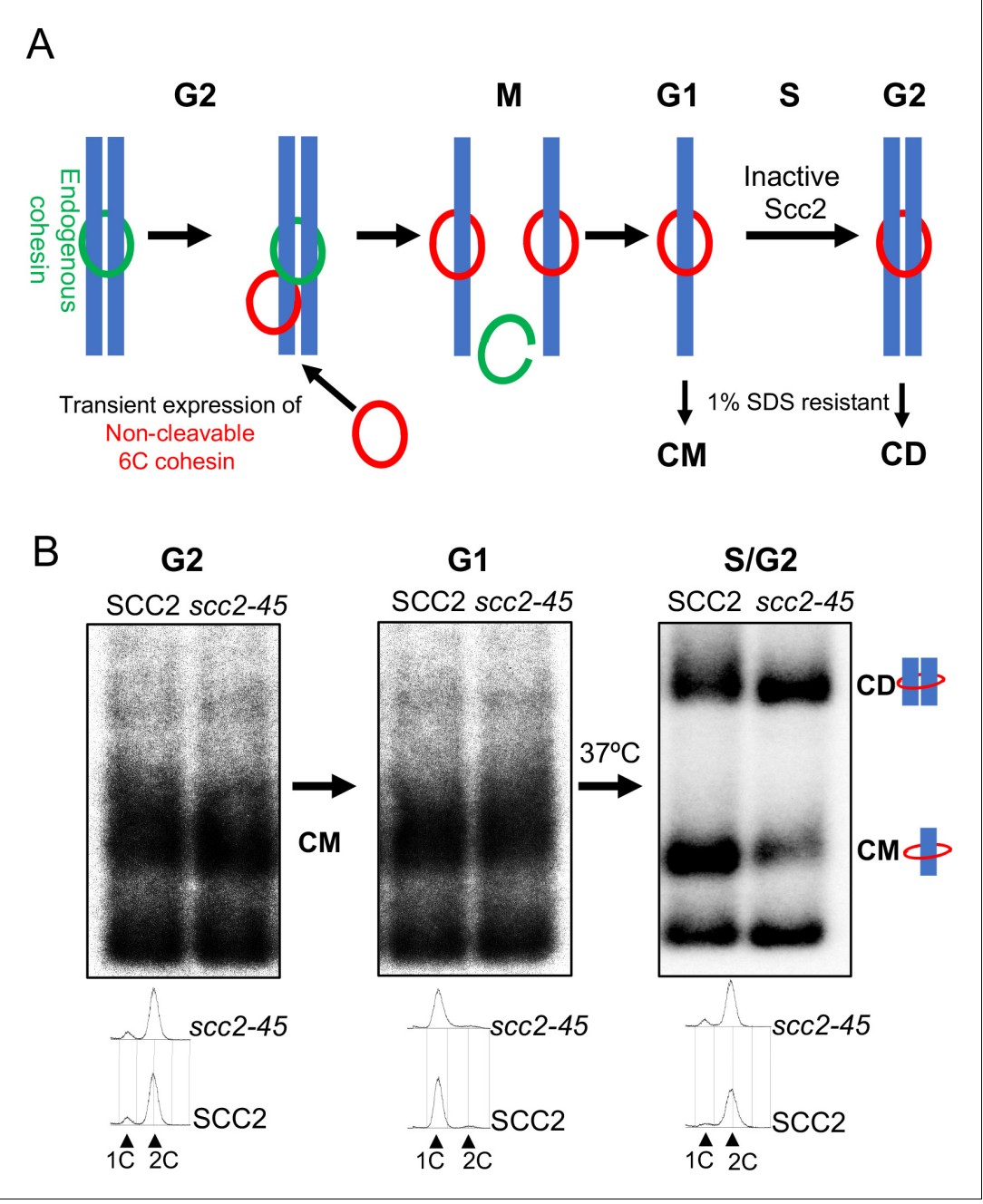

**Figure 2.** Chromosomal cohesin is converted into cohesive structures in the absence of Scc2 activity. (**A**) A schematic depiction of the cohesin conversion assay, see the results and Materials and methods sections for detailed description of the assay. Green rings denote endogenous cohesin and red rings denote non-cleavable 6C cohesin. Only the latter molecules are capable of generating SDS-resistant catenated monomers (CMs) and catenated dimers (CDs). (**B**) Wild type (K24697) and *scc2-45* (temperature sensitive mutant of Scc2) (K24738) strains that contain genes coding for 6C non cleavable cohesin (2C *SMC1* 2C *SMC3* and *GALp*-2C *SCC1*^NC) were arrested in G2 phase, the expression of 2C Scc1^NC was induced by addition of galactose for 45 min. Mini-chromosome IP of the cultures at this stage shows CMs formed in both strains. The cultures were released from the G2 arrest and arrested in the subsequent the G1 phase, mini-chromosome IP of the cultures at this stage shows CMs formed in the previous G2 phase being retained in the subsequent G1 in both strains. The cultures were released from the G1 arrest and allowed to undergo replication at 37˚C (in order to inactivate Scc2 in the *scc2-45* strain), mini-chromosome IP shows formation of CDs in both the wild type and *scc2-45* strains, in the *scc2-45* mutant strain this is accompanied by a reduction in the amount of CMs. The FACS profiles of the two cultures at different stages of

*Figure 2 continued on next page*

*Figure 2 continued*

the experiment is shown below the respective southern blots. The data shown is representative of three independent biological repeats.

The online version of this article includes the following figure supplement(s) for figure 2:

**Figure supplement 1.** Non cleavable cohesin expressed in the G2 phase survives mitosis and remains stably associated with the chromosomes in the subsequent G1 phase.

formation in otherwise wild-type cells; that is, when cohesion can in principle be generated by de novo cohesin loading as well as conversion. To do this, we synchronised wild type, *mrc1Δ*, *chl1Δ*, *ctf4Δ*, *csm3Δ*, *tof1Δ* and *ctf18Δ* cells expressing endogenous 6C cohesin in G1 and released them into a G2/M arrest and measured formation of CMs and CDs. As expected, none of the deletions affected CMs but more remarkably none caused any appreciable reduction in CDs either (*Figure 3A*). When cohesion of an individual chromosomal locus marked by GFP is measured, all six deletions cause modest but reproducible cohesion defects (*Borges et al., 2013*; *Xu et al., 2007*), suggesting that the 6C mini-chromosome assay is less sensitive to minor cohesion establishment defects than GFP-based cytological assays. We suggest that this difference may arise because mini-chromosomes are merely 2.3 kb in length and contain *CEN*s that act as potent cohesin loading sites (*Hu et al., 2015*). Crucially, cells lacking individual members of the above set of non-essential replisome associated proteins are perfectly capable of generating and maintaining cohesion of the mini-chromosomes, as measured by CD formation, when Scc1 expressed from its endogenous promoter can be loaded throughout G1 and S phase.

To measure the role of these proteins in cohesin conversion, we transiently induced during a nocodazole-induced G2/M arrest 2CScc1$^{NC}$ cohesin in wild type, *mrc1Δ*, *chl1Δ*, *ctf4Δ*, *csm3Δ*, *tof1Δ* and *ctf18Δ* cells expressing 2CSmc1 and 2CSmc3 and then released them from nocadazole into a pheromone-induced G1 arrest. In all six mutants as well as wild type, 2CScc1$^{NC}$ cohesin produced during G2/M cohesin survived anaphase and remained associated with DNA during the G1 arrest, with a substantial fraction in the form of CMs (*Figure 3B*).

When allowed to undergo DNA replication upon release from the pheromone-induced G1 arrest (*Figure 3—figure supplement 1*), *ctf18Δ* and *mrc1Δ* cells formed CDs in a manner similar if not identical to wild type (*Figure 3B*). They also contained a substantial level of CMs and free supercoiled monomers merely associated with 2CScc1$^{NC}$. Remarkably, the behaviour of *chl1Δ*, *ctf4Δ*, *csm3Δ*, and *tof1Δ* cells was completely different. No CDs were produced and most but not all CMs disappeared, as did most monomeric supercoils associated in a non-topological fashion with 2CScc1$^{NC}$ cohesin (*Figure 3B*). These data suggest that Chl1, Ctf4, and Csm3/Tof1 (but not Ctf18-RFC) are essential for converting cohesin associated with un-replicated mini-chromosomes into cohesive structures as measured by CDs. Equally surprising, these replisome proteins are also necessary for preventing cohesin's dissociation from chromosomes during the passage from G1 to G2. It is remarkable that their division into a group (containing Chl1, Ctf4, and Csm3/Tof1) required for conversion and one (containing Mrc1 and Ctf18-RFC) that is not, corresponds precisely with the two epistasis groups defined by the effects of double mutations on cohesion and viability (*Xu et al., 2007*). Our data suggest that Chl1, Ctf4, Csm3/Tof1 form a single epistasis group because they are all concerned with a very specific physiological function, namely conversion of pre-existing chromosomal cohesin into cohesive structures. An important corollary of our finding that Chl1, Ctf4, Csm3/Tof1 are essential for conversion (*Figure 3B*) but not for building cohesion per se (*Figure 3A*) or for cell proliferation, is that conversion is not the sole mechanism by which cohesion is established during S phase. As implied by the synthetic lethality caused by combining mutants from different epistasis groups (*Xu et al., 2007*) and as will become more apparent below, a second mechanism involving Mrc1 and Ctf18-RFC is capable of generating cohesion in the complete absence of conversion.

## Cohesin associated with un-replicated DNA is evicted from mini-chromosomes during S phase in the absence of the conversion pathway

Although no CDs are formed and few CMs retained when *chl1Δ*, *ctf4Δ*, *csm3Δ*, and *tof1Δ* mutants undergo S phase, a low but significant level of CMs can nevertheless be detected in the mutant cells once they reach G2/M (*Figure 3B*). These could either represent rare chromosomal cohesin

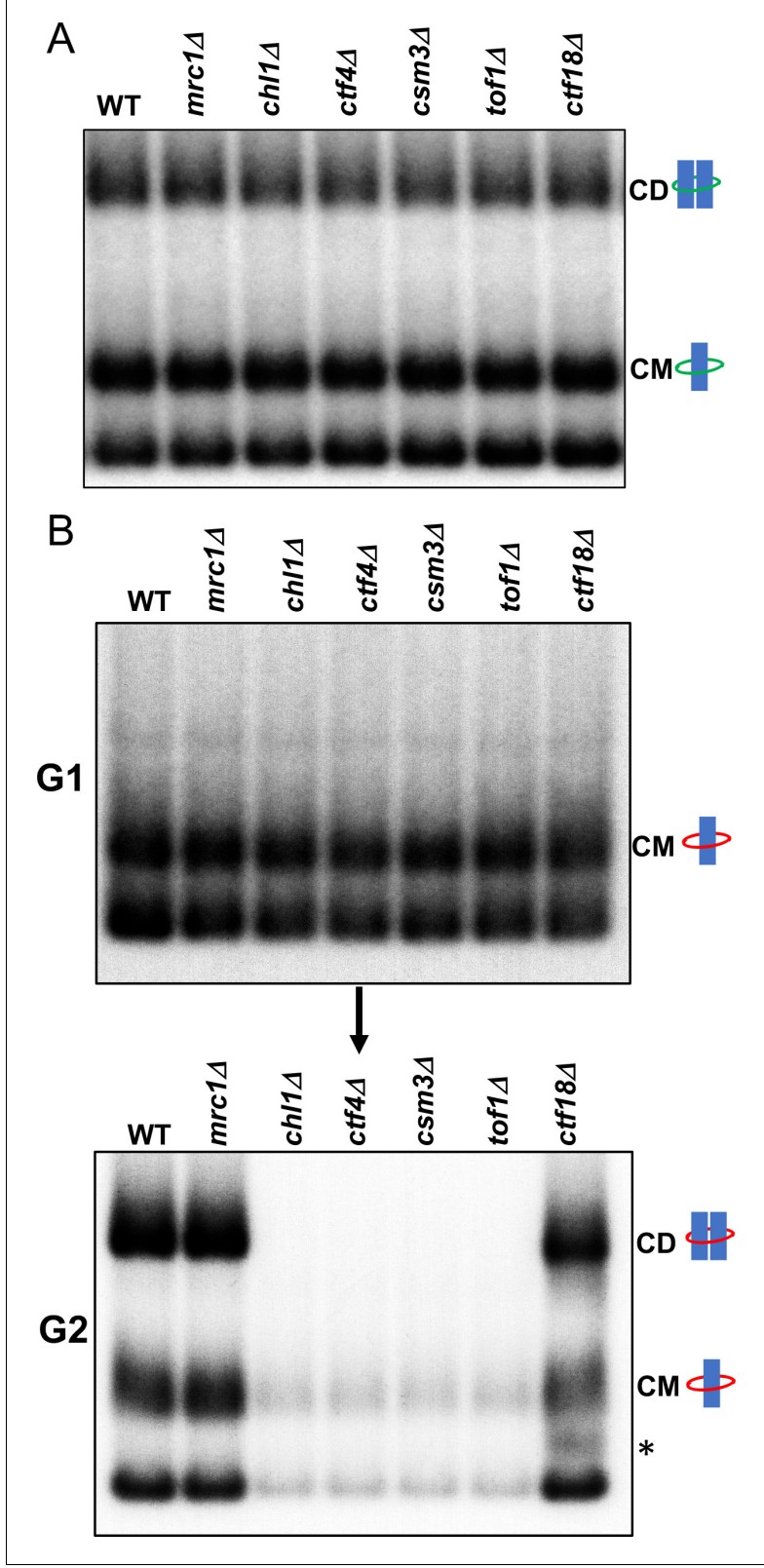

**Figure 3.** Tof1/Csm3, Ctf4 and Chl1 are essential for cohesin conversion. (**A**) Exponentially growing wild type (K23889), *mrc1Δ* (K28092), *chl1Δ* (K28082), *ctf4Δ* (K28084), *csm3Δ* (K28108), *tof1Δ* (K28091) and *ctf18Δ* (K28115) expressing endogenous 6C cohesin were synchronised in G1, released into a G2/M arrest and subjected to in vivo crosslinking and mini-chromosome IP. The CM and CD bands are marked. Data shown is representative of two

*Figure 3 continued on next page*

*Figure 3 continued*

independent biological repeats. (B) Wild type (K24697) *mrc1Δ* (K28276), *chl1Δ* (K28175), *ctf4Δ* (K28275), *csm3Δ* (K28282), *tof1Δ* (K28280) and *ctf18Δ* (K28285) strains that contain genes coding for 6C non cleavable cohesin (2C *SMC1* 2C *SMC3* and *GALp*-2C *SCC1*^NC) were arrested in G2 phase, the expression of 2C Scc1^NC was induced by addition of galactose for 45 min. The cultures were released from the G2 arrest and arrested in the subsequent the G1 phase, mini-chromosome IP of the cultures at this stage shows CMs formed in the previous G2 phase being retained in the subsequent G1 in all the strains. The cultures were released from the G1 arrest and allowed to undergo replication and subjected to mini-chromosome IP, the CM and CD bands are marked. A species migrating between the CMs and supercoiled mini-chromosome DNA routinely detected in the *ctf18Δ* strain is marked with *. We currently do not know the identity of this band. Data shown is representative of two independent biological repeats.

The online version of this article includes the following figure supplement(s) for figure 3:

**Figure supplement 1.** FACS profiles of the cultures described in *Figure 3B*.

---

complexes that survive replication in the absence of conversion proteins or complexes that reload onto chromosomes following their eviction during DNA replication. To distinguish these possibilities, we analysed conversion in wild-type *SCC2*, *scc2-45*, *SCC2 chl1Δ*, and *scc2-45 chl1Δ* cells. When constructing these strains, we found that combining *scc2-45* with *chl1Δ* caused synthetic sickness (*Figure 4—figure supplement 1A*). However, this did not prevent us from performing experiments with the double mutant strain. In this case, 2CScc1^NC cohesin was loaded in G2/M cells at 25°C and cells shifted to 37°C 20 min before release from their pheromone-induced G1 arrest. As expected, both wild type and *scc2-45* cells produced CDs from 2CScc1^NC cohesin loaded during the previous G2/M while neither *chl1Δ*, nor *scc2-45 chl1Δ* cells did so (*Figure 4A*). Crucially, a small fraction of CMs and free supercoiled monomers associated with 2CScc1^NC cohesin persisted in *chl1Δ* but none in *scc2-45 chl1Δ* cells (*Figure 4A*). The failure to detect either type of interaction after *scc2-45 chl1Δ* cells undergo S phase at 37°C was not due to reduced stability of Scc1^NC protein or mini-chromosome DNA as their levels were similar in wild-type *SCC2*, *scc2-25*, *SCC2 chl1Δ*, and *scc2-45 chl1Δ* cells (*Figure 4—figure supplement 1B and C*). We conclude that Chl1, Ctf4, and Csm3/Tof1 are important for maintaining cohesin on mini-chromosome DNA during S phase and that the low levels that do persist (at least in *chl1Δ* cells) arise from Scc2-dependent re-loading, either during or following passage through S phase.

## Cohesin associated with un-replicated DNA is evicted from the entire genome when *chl1Δ* cells undergo S phase

To address whether the eviction of chromosomal cohesin when *chl1Δ* mutants undergo S phase is limited to mini-chromosomes, we repeated the experiment and measured the fate during S phase of cohesin associated with un-replicated DNA using calibrated ChIP sequencing (*Figure 4—figure supplement 2*; *Hu et al., 2015*). Consistent with previous observations (*Srinivasan et al., 2019*), Scc1^NC cohesin (loaded during the previous G2/M at 25°C) persisted throughout the genome during passage from G1 to G2 at 37°C in both wild type and *scc2-45* cells. (*Figure 4B*). This merely confirms that though normally required to maintain cohesin on chromosomes during G1, Scc2 is not necessary when the chromosomal cohesin had been loaded during the previous G2 period and is refractory to separase cleavage. In contrast, half or more Scc1^NC cohesin (loaded during the previous G2/M at 25°C) disappeared from all parts of genome when *chl1Δ* cells underwent S phase (*Figure 4B*). This effect that was even more pronounced in *chl1Δ scc2-45* double mutant cells, which cannot reload cohesin during S phase, where nearly all pre-existing chromosomal Scc1^NC cohesin disappeared during the passage from G1 to G2 (*Figure 4B*). We conclude that in addition to being required to build CDs from cohesin associated with un-replicated DNAs, Chl1 is essential for maintaining cohesin's association with the entire genome as cells undergo S phase.

## Cohesion can also be established by Scc2-dependent de novo loading of nucleoplasmic cohesin

*chl1Δ*, *ctf4Δ*, *csm3Δ*, and *tof1Δ* cells are incapable of generating CDs by converting non-cleavable cohesin associated with un-replicated DNAs but have no such defect when wild-type cohesin is expressed from its endogenous promoter during late G1 and S phase (*Figure 3A and B*). To address

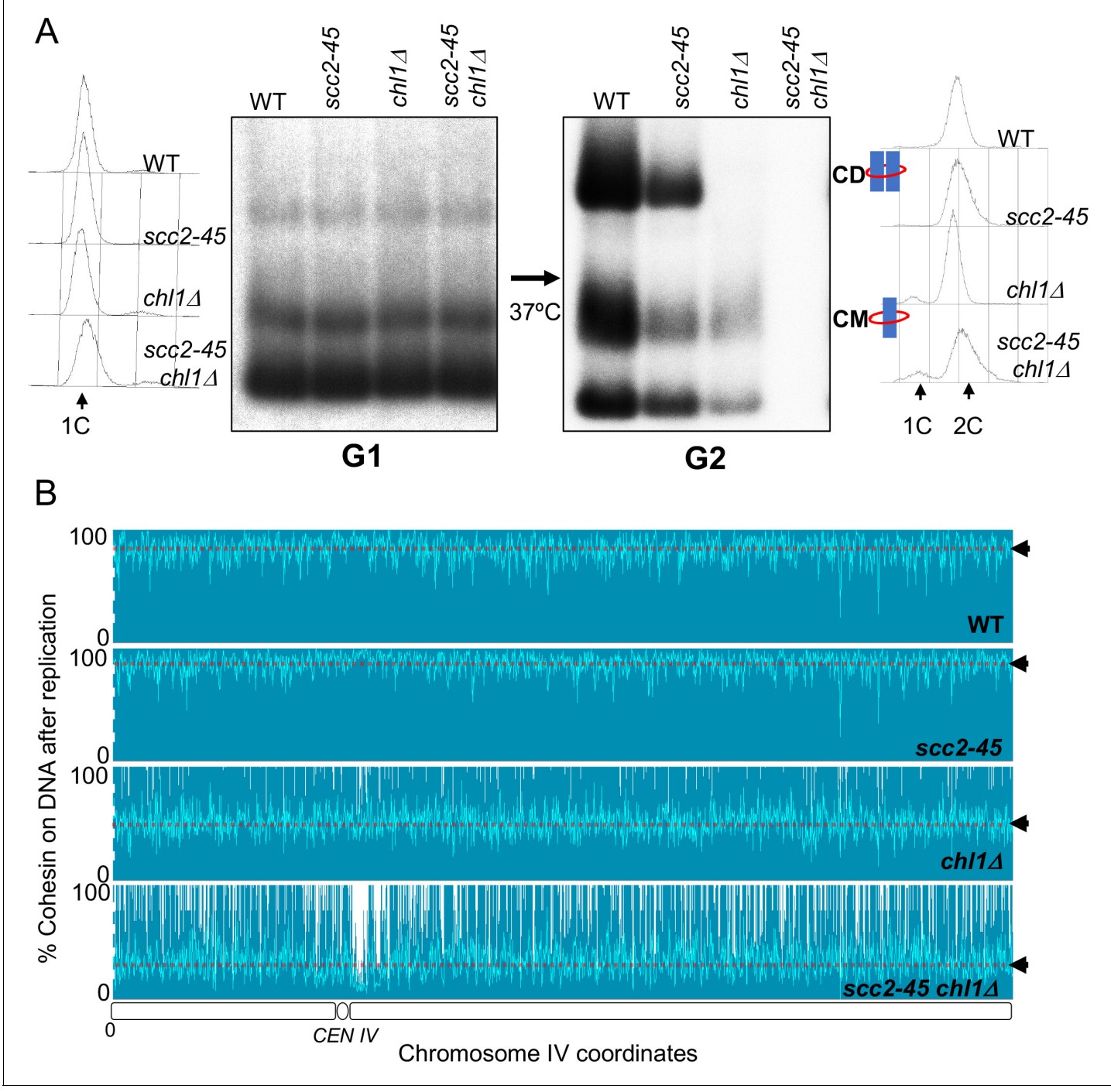

**Figure 4.** Chromosome associated cohesin is evicted during S phase in the absence of the conversion pathway. (**A**) Wild type (K24697), *scc2-45* (K24738), *chl1Δ* (K28175) and *chlΔ scc2-45* (K28061) strains that contain genes coding for 6C non cleavable cohesin (2C *SMC1* 2C *SMC3* and *GALp*-2C *SCC1*[NC]) were arrested in G2 phase, the expression of 2C Scc1[NC] was induced by addition of galactose for 45 min. The cultures were released from the G2 arrest and arrested in the subsequent the G1 phase, mini-chromosome IP of the cultures at this stage shows CMs formed in the previous G2 phase being retained in the subsequent G1 in all the strains. The cultures were released from the G1 arrest and allowed to undergo replication at 37°C (in order to inactivate Scc2 in the *scc2-45* strain). Mini-chromosome IP shows formation of CDs in both the wild type, and *scc2-45* strains, the *chl1Δ* strain did not form CD and showed a reduction in CM level and the *chlΔ scc2-45* strain failed to IP any mini-chromosome DNA. The FACS profiles of the cultures at different stages of the experiment are shown next to the respective southern blots. Data shown is representative of two independent biological repeats. (**B**) Samples drawn from the cultures described above during the G1 phase and after replication at 37°C were subjected to calibrated ChIP sequencing with anti-PK antibody to measure Scc1[NC] levels (See *Figure 4—figure supplement 2* for the ChIP profiles). The change in Scc1[NC] levels on chromosome IV after replication is shown, median level of Scc1[NC] across the entire chromosome IV (dotted line) is marked with arrowheads.

*Figure 4 continued on next page*

*Figure 4 continued*

The online version of this article includes the following figure supplement(s) for figure 4:

**Figure supplement 1.** Chromosome associated cohesin is evicted during S phase in the absence of the conversion pathway.

**Figure supplement 2.** The occupancy of Scc1$^{NC}$ along chromosome IV in the experiment described in *Figure 4B*.

whether this discrepancy could be an artefact caused by the mutations that prevent 2CScc1$^{NC}$'s cleavage, we synchronised wild type, *chl1Δ*, *ctf4Δ*, *csm3Δ*, and *tof1Δ* cells in G1 and then allowed them to undergo S phase in the presence of galactose to induce 2CScc1$^{NC}$. Under these circumstances, CD formation was identical in wild type and mutant cells (*Figure 5A*), proving that it is the time of synthesis and not whether it is non-cleavable that determines whether Scc1 can generate CDs in *chl1Δ*, *ctf4Δ*, *csm3Δ*, and *tof1Δ* mutant cells. When expressed during S phase, 2CScc1$^{NC}$ can generate CDs but not when expressed only transiently during the previous G2/M phase. In the former case, cohesin is free to load de novo during S phase while in the latter it must be converted from cohesin already associated with mini-chromosome DNA.

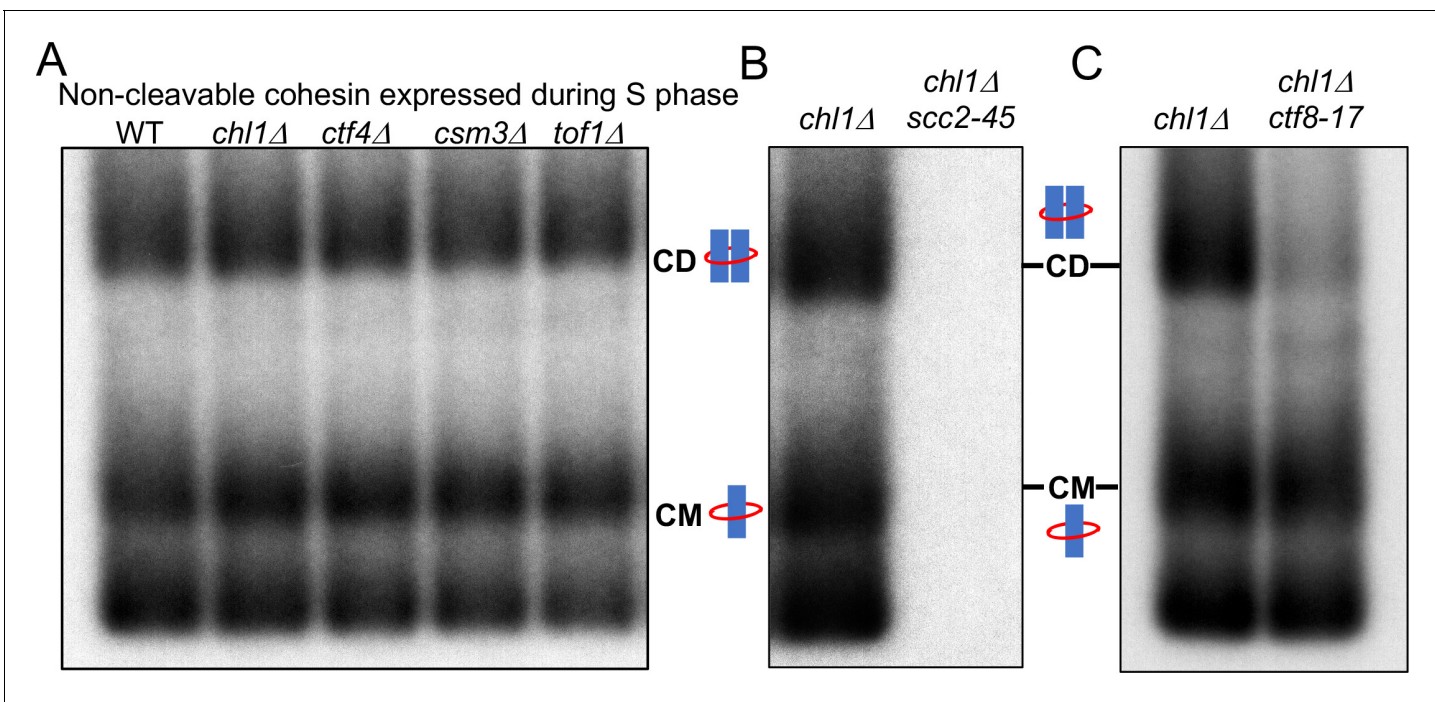

**Figure 5.** Cohesion can also be established by CTF18-RFC and Scc2 dependent de novo loading of nucleoplasmic cohesin. (**A**) Wild type (K24697), *chl1Δ* (K28175), *ctf4Δ* (K28275), *csm3Δ* (K28282) and *tof1Δ* (K28280) strains that contain genes coding for 6C non cleavable cohesin (2C *SMC1* 2C *SMC3* and *GALp*-2C *SCC1$^{NC}$*) were arrested in G1 phase, the expression of 2C Scc1$^{NC}$ was induced by addition of galactose for 15 min. The cultures were released from the G1 arrest into medium containing galactose in order to ensure constant expression of Scc1$^{NC}$ during DNA replication, mini-chromosome IP of the cultures after replication shows CM and CD formation in all the strains. Data shown is representative of two independent biological repeats. (**B**) *chl1Δ* (K28175) and *chl1Δ scc2-45* (K28061) strains that contain genes coding for 6C non-cleavable cohesin (2C *SMC1* 2C *SMC3* and *GALp*-2C *SCC1$^{NC}$*) were arrested in G1 phase, the expression of 2C Scc1$^{NC}$ was induced by addition of galactose for 15 min. The cultures were released from the G1 arrest into medium containing galactose in order to ensure constant expression of Scc1$^{NC}$ during DNA replication at 37°C (in order to inactivate Scc2 in the *scc2-45* strain). Mini-chromosome IP of the cultures after replication shows CM and CD formation in the *chl1Δ* while in the *chl1Δ scc2-45* we failed to IP any mini-chromosome DNA. Data shown is representative of two independent biological repeats. (**C**) *chl1Δ* (K28175) and *chl1Δ ctf8-17* (K28295) strains that contain genes coding for 6C non cleavable cohesin (2C *SMC1* 2C *SMC3* and *GALp*-2C *SCC1$^{NC}$*) were arrested in G1 phase, the expression of 2C Scc1$^{NC}$ was induced by addition of galactose for 15 min. The cultures were released from the G1 arrest into medium containing galactose in order to ensure constant expression of Scc1$^{NC}$ during DNA replication at 37°C (in order to inactivate Ctf8 in the *ctf8-17* strain). Mini-chromosome IP of the cultures after replication shows CM and CD formation in the *chl1Δ*. While the *chl1Δ ctf8-17* strains showed the presence of CMs, the level of CDs was greatly reduced. Data shown is representative of two independent biological repeats.

The online version of this article includes the following figure supplement(s) for figure 5:

**Figure supplement 1.** Cohesion can also be established by CTF18-RFC and Scc2 dependent de novo loading of nucleoplasmic cohesin.

These data along with previous observations (*Xu et al., 2007*) suggest that a second mechanism establishes sister chromatid cohesion during S phase, namely one that involves de novo loading during S phase, which acts in parallel and independently of one that converts cohesin associated with un-replicated DNA. Because both mechanisms are sufficient to generate cohesion, especially in the case of mini-chromosome CDs (see *Figure 5A*), we set out to characterize the de novo pathway by measuring cohesion establishment in cells lacking the conversion pathway, namely *chl1Δ* cells. Our first question concerned Scc2, which should be necessary for cohesion established from nucleoplasmic cohesin. To address this, *chl1Δ* and *chl1Δ scc2-45* cells were synchronised in G1 at 25°C and then allowed to undergo replication at 37°C in the presence of galactose to ensure continuous synthesis of 2CScc1$^{NC}$ from the *GAL* promoter throughout S phase. As expected, *chl1Δ* cells produced both CMs and CDs. In contrast, neither CDs nor CMs nor any other form of interaction between 2CScc1$^{NC}$ cohesin and DNA was observed in *chl1Δ scc2-45* cells (*Figure 5B*). Importantly, this absence was not due to reduced Scc1 accumulation or due to disappearance of the mini-chromosome DNA (*Figure 5—figure supplement 1A and B*). We conclude that the de novo pathway requires Scc2, which is not particularly surprising as it has long been known that Scc2 is required for endogenous nucleoplasmic cohesin to load onto chromosomes when G1 cells undergo S phase (*Ciosk et al., 2000*).

## The de novo pathway requires RFC-Ctf18

Though *ctf18Δ* and *mrc1Δ* mutant cells form CDs by converting cohesin associated with un-replicated DNA with an efficiency comparable to wild type (*Figure 3B*), both mutations cause significant cohesion defects when measured using a cytological assay (*Borges et al., 2013*; *Xu et al., 2007*). They are also synthetic lethal with *chl1Δ*, *ctf4Δ*, *csm3Δ*, or *tof1Δ* . Indeed, we confirmed by tetrad analysis that deleting *MRC1* or *CTF18* in *chl1Δ* mutants does indeed cause extreme sickness or lethality (*Figure 5—figure supplement 1C*). Due to this lethality, it is not possible to measure the effect on CD formation by the de novo pathway by deleting *MRC1* or *CTF18*. We therefore addressed the issue using the conditional *ctf8-17* mutation that inactivates the Ctf8 subunit of the Ctf18-RFC complex when cells are shifted to 37° (*Xu et al., 2007*). Because the Ctf18-RFC is not essential for viability, *ctf8-17* does not abrogate proliferation at the non-permissive temperature but does so when combined with *chl1Δ*, *ctf4Δ*, *csm3Δ*, or *tof1Δ* (*Xu et al., 2007*).

To measure the effect on the de novo pathway of inactivating Ctf18-RFC using the temperature sensitive *ctf8-17* allele, we synchronised *chl1Δ* and *chl1Δ ctf8-17* double mutant cells (expressing 2CSmc1 and 2CSmc3) in G1 at the permissive temperature (25°C) and released them into a G2/M arrest at 37°C in the presence of galactose (to induce the expression of 2CScc1$^{NC}$). CDs were formed efficiently in *chl1Δ* cells but barely if at all in the *chl1Δ ctf8-17* double mutants (*Figure 5C*) despite their completing DNA replication (*Figure 5—figure supplement 1E*). Although CD formation was abrogated by the *ctf8-17* mutation, CM formation was unaffected, as were all forms of association between cohesin and mini-chromosome DNA, implying that Scc2 remains active in the *chl1Δ ctf8-17* double mutant cells and loads cohesin onto individual DNA molecules either during or after replication (*Figure 5C*). Crucially, it cannot load cohesin onto nascent DNAs in a manner that entraps them inside the same cohesin ring. We conclude that establishment of cohesion by the de novo pathway depends not only on Scc2 but also on the Ctf18-RFC complex. A corollary is that there are only two pathways capable of creating CDs during S phase and inactivation of both causes a loss of cohesion so complete that it causes lethality (*Xu et al., 2007*).

## Over-expression of *CHL1* suppresses cohesion defects caused by the de novo pathway *ctf8Δ* mutation

Although not lethal in mitotic cells, *ctf8Δ* causes spore inviability due to massive chromosome non-disjunction during the second meiotic division, as do mutations of conversion pathway proteins (*Petronczki et al., 2004*). Interestingly, *CHL1* was identified as a multicopy suppressor of the poor spore viability of *ctf8Δ* cells (*Petronczki et al., 2004*), suggesting that it may be the rate-limiting factor of the conversion pathway and that the augmentation of conversion pathway by Chl1 over-expression can compensate for defects in the de novo pathway (due to *ctf8Δ*), at least during meiosis.

To address whether this also applies to mitotic cells and to test whether the suppression of cohesion defects by Chl1 over-expression is specific to cells lacking the de novo pathway, we used a cytological assay (*Petronczki et al., 2004*) to compare the effect of multicopy *CHL1* on the precocious separation of sister chromatids in cycling *ctf8Δ* and *ctf4Δ* mutants. We visualized sister-chromatid cohesion at the *URA3* locus (*URA3-GFP*) by expressing a tet-repressor–GFP fusion protein in wild type, *ctf8Δ* and *ctf4Δ* strains expressing a Myc-tagged version of Securin (Pds1-Myc18) and carrying a tandem array of tetracycline operators integrated at the *URA3* locus 35 kb away from the centromere of chromosome V. Each strain contained either an empty 2-micron multicopy vector or the 2-micron episome containing the *CHL1* (+*CHL1*) gene. Cycling cells were fixed and stained with DAPI to visualize DNA and antibodies against the Myc epitope to evaluate securin levels. We identified metaphase cells on the basis of their large buds and high levels of nuclear securin and determined the proportion of these cells containing one or two *URA3-GFP* signals. To facilitate this analysis, all strains were diploids, which have a larger cell size, but crucially only a single copy of the *URA3* locus was marked by GFP.

In wild-type cells, sister chromatid disjunction reveals two separate GFP dots, an event that only takes place once securin has been destroyed by the APC/C. Precocious loss of sister chromatid cohesion is revealed by the appearance of two separate *URA-GFP* dots in large budded cells whose nuclei have high levels of securin and in cells containing the two micron vector alone this occurred in 49% of *ctf4Δ* and 47% of *ctf8Δ* cells (*Figure 6A*). In contrast, introduction of *CHL1* on the 2-micron vector caused a marked reduction from 47% to 18% in *ctf8Δ* cells but a much more modest effect, from 49% to 40%, in *ctf4Δ* cells (*Figure 6A*). These data suggest that augmentation of the conversion pathway by *CHL1* over-expression can compensate for defects in the de novo pathway in mitotic cells as well as meiotic ones. Importantly, it does not compensate for defective conversion caused by *ctf4Δ*. Because Ctf4 remains important for conversion even when *CHL1* is over-expressed, the ability of Chl1 to promote conversion depends on Ctf4's association with the replisome, which is consistent with the suggestion that Chl1's association with chromatin depends on Ctf4 (*Samora et al., 2016*).

## Duplication of the *SCC2* suppresses the proliferative defects of *ctf4Δ* but not *ctf18Δ* mutants

Cells lacking Ctf4 display reduced fitness relative to wildtype cells. The difference in fitness is measured by competing cells lacking Ctf4 with a reference, wild-type, strain and measuring their relative abundance over several generations (*Fumasoni and Murray, 2020*). It was recently reported that the reduced fitness of *ctf4Δ* relative to the wildtype cells is partially suppressed by duplication of two different portions of the genome that contain the *SCC2* and *SCC4* genes. Indeed, an extra copy merely of *SCC2* partially ameliorates their cohesion defect (*Fumasoni and Murray, 2020*). This finding implies not only that the cohesion defect contributes to the reduced fitness but also that increased Scc2/4 activity enhances establishment of cohesion by a Ctf4-independent pathway, presumably the de novo loading pathway dependent on Ctf18-RFC. Unlike the conversion pathway, which can use cohesin molecules previously loaded onto chromosomes, cohesion created by the de novo pathway may have only a very narrow time window within which cohesin must be loaded. If loaded significantly before replication, then the cohesin is presumably converted but if loaded too late after replication, it may be unable to generate cohesion. For example, cohesin loaded during G2 produces CMs but not CDs (*Srinivasan et al., 2019*). If de novo loading cohesion establishment takes place at replication forks and involves the creation of cohesive structures between double-stranded leading strands and single-stranded lagging strands (*Murayama et al., 2018*), then the window of opportunity may be very short indeed. Because conversion does not require Scc2 (except to load cohesin in the first place at any stage preceding replication) while the de novo pathway does and has a narrow window of opportunity, the latter may be particularly sensitive to the levels of Scc2/4 at replication forks. According to this logic, the modest overexpression of Scc2 that appears to enhance cohesion establishment by Ctf18-RFC (in *ctf4Δ* cells) should have less or even no effect on cohesion established by the conversion pathway (in *ctf18Δ* cells). If on the other hand, there were no major mechanistic difference between cohesion created by Ctf4 and Ctf18, then *SCC2* duplication might suppress the cohesion defects of *ctf4Δ* cells merely because it increases the amount of cohesin on chromosomes. In which case, *SCC2* duplication should ameliorate the reduced fitness of *ctf18Δ* (*Figure 6—figure supplement 1*) as well as *ctf4Δ* (*Figure 6—figure supplement 1*) mutants.

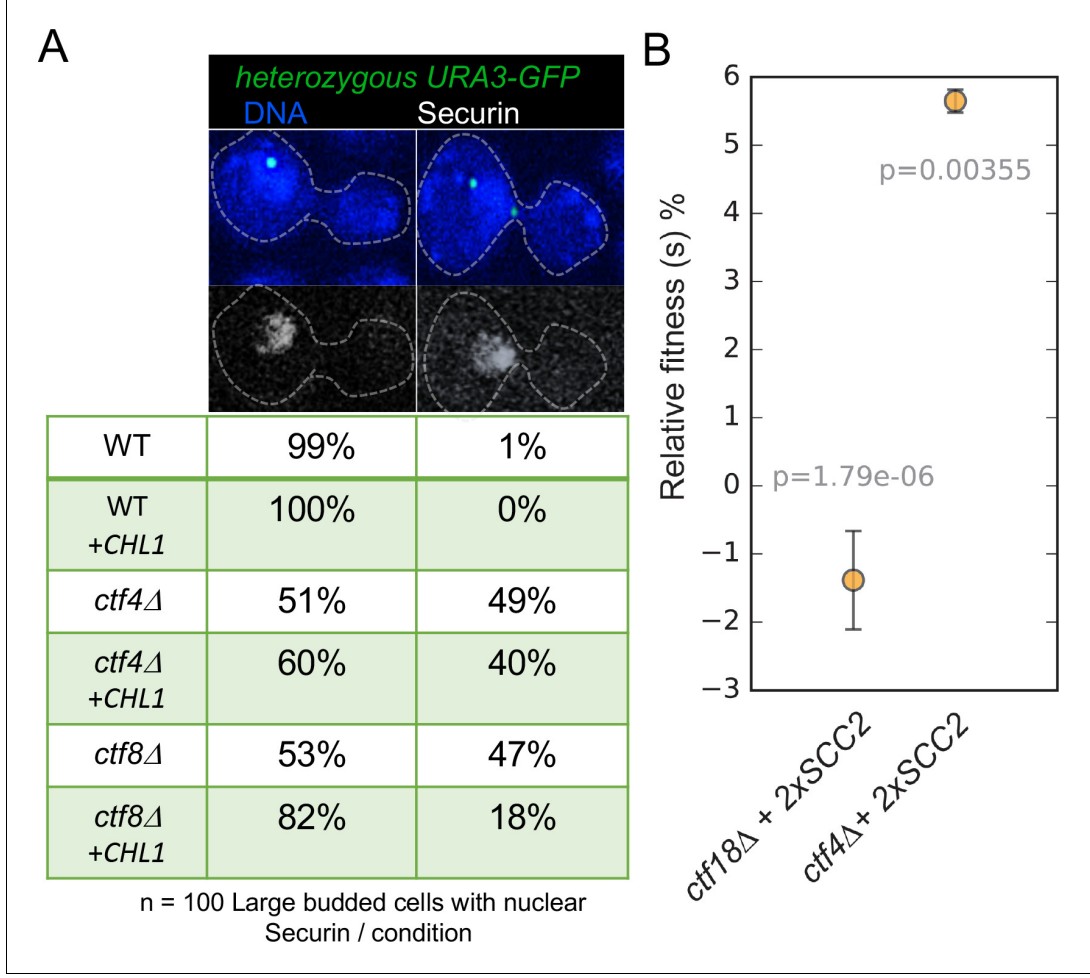

**Figure 6.** Augmentation of either cohesion establishment pathway suppresses the cohesion defects caused by abrogation of the other. (**A**) Asynchronous cultures of wild-type (K10003), *ctf4Δ* (K11692), *ctf8Δ* (K10349) that carry either an empty 2μ vector or a 2μ vector containing the *CHL1* gene were fixed and stained for DNA and with an antibody against the Myc epitope as detailed in Materials and methods. The strains express Pds1-Myc18 (securin) and contain chromosome V marked by GFP at the *URA3* locus, 35 kb away from the centromere. Fluorescence images of representative large budded metaphase cell showing one or two *URA3*-GFP dots (green), nuclear securin (white) and DNA (blue) is shown. The proportion of securin-positive cells containing one or two *URA3-GFP* signals in the different strains is listed below the images. 100 metaphase cells were scored for each strain in this experiment. (**B**) The fitness of ancestral *ctf4Δ* and *ctf18Δ* strains carrying an extra copy of the SCC2 gene, relative to the *ctf4Δ* and *ctf18Δ* ancestors respectively. Error bars represent standard deviation, the p values from unequal variances t-tests are also shown. The source data is presented in *Source data 1*.

The online version of this article includes the following figure supplement(s) for figure 6:

**Figure supplement 1.** The fitness of wild type, *ctf18Δ* and *ctf4Δ* strains relative to the wild-type ancestor.

To test this, we competed strains carrying deletion of either *CTF4* or *CTF18* with respective *ctf4Δ* and *ctf18Δ* strains that carry an additional copy of *SCC2* expressed under its native promoter and integrated at the *URA3* locus. The relative fitness of the strains carrying the extra copy of *SCC2* was calculated by measuring the ratios of the two competing genotypes over 30 generations (see Materials and methods for details). Crucially, an additional copy of *SCC2* increased the fitness of *ctf4Δ* (*Figure 6B* and *Source data 1*) but not that of *ctf18Δ* cells (*Figure 6B* and *Source data 1*). If anything, the latter was modestly reduced. This is consistent with the notion that increased Scc2/4 levels suppresses *ctf4Δ* because it specifically enhances the Scc2-dependent de novo loading cohesion establishment pathway.

Thus, establishment of cohesion via the de novo loading pathway but not that of conversion is specifically augmented by modest over-expression of Scc2 while cohesion established by the conversion pathway is augmented by *CHL1* over-expression. Importantly, augmentation of either pathway in this manner suppresses the reduction in cohesion establishment caused by abrogation of the other.

## Discussion

To understand the mechanism by which sister DNAs are joined together by cohesin, it is necessary to know when precisely during the cell cycle this occurs. It is self-evident that if sister DNAs were to disjoin fully, as eventually happens in cohesin mutants, then it would be difficult if not impossible to ensure that sister DNAs were joined together in register. Even if sequence-specific pairing were possible, how would cells distinguish sisters from homologous non-sister DNAs? At first sight, it might therefore appear obvious that cohesion must be established as early as possible after sisters are created. However, due largely to rotation of the replisome during fork progression, nascent sister DNAs become entwined around each other as well as being held together by cohesin (*Farcas et al., 2011*). Much of this entanglement is rapidly removed by Topoisomerase II but some persists, albeit in a manner partially dependent on cohesin, until mitosis. Because of this DNA-DNA catenation, sister DNAs could in principle remain close enough to each other even during G2 for cohesin to recognize them as such and thereby generate cohesion long after the passage of replication forks. In fact, this does not occur because expression of Scc1 exclusively after S phase cannot generate sister chromatid cohesion, at least in yeast cells (*Uhlmann and Nasmyth, 1998*). There are two explanations for this finding. Either sister DNAs are insufficiently close together under these conditions or the cohesion establishment mechanism simply cannot function once cells enter G2, possibly because the process requires replisome proteins or DNA features specific to nascent DNA, like one sister needing to be single stranded. This raises the question whether cohesin can build new cohesive structures during G2 even when sisters are already held together by cohesin. The fact that this also does not take place even though cohesin can be loaded onto chromosomes during this stage of the cell cycle implies that cohesion establishment activity is confined to S phase (*Haering et al., 2004*).

If confined to S phase, does cohesion establishment involve proteins associated with replisomes? The problem with addressing such a question is that most replisome proteins are essential for DNA replication, without which no cohesion is possible. As it happens, the collection of known replisome proteins includes numerous members that are not in fact necessary either for replication or even for proliferation. Amongst these are a subset whose abrogation causes significant defects in sister chromatid cohesion, albeit more modest than that caused by cohesin's inactivation. Examples include Mrc1, Chl1, the Tof1/Csm3 complex, and Ctf4, which are associated with replication forks, and a variant RFC in which Rfc1 is replaced by Ctf18 bound by Ctf8 and Dcc1 (Ctf18-RFC). An extensive analysis of double mutants led to the suggestion that their genes fall into two epistasis groups, one containing *TOF1/CSM3*, *CHL1*, and *CTF4* while the other *MRC1* and *CTF18*.

The involvement of replisome-associated proteins implied that cohesion is indeed established at replication forks. Moreover, the existence of two different epistasis groups explained why none of these proteins are essential. If the two groups were involved in different mechanisms, each capable of establishing sufficient cohesion for cell proliferation, then lethality would only arise if both mechanisms were abrogated. However, hitherto there has been no rationale for why two mechanisms exist or what physiological differences distinguish them. In this paper, we describe a fresh approach to this conundrum. We started with a very different type of question, namely whether or not cohesion is established from cohesin previously associated with un-replicated DNAs, in a process that converts the latter directly into cohesive structures without dissociating from DNA.

Having developed a method to follow the fate of chromosomal cohesin, we show that it is indeed converted into cohesive structures during S phase, that this process does not require Scc2 (except to load cohesin in the first place), but that it is strictly dependent on Tof1, Csm3, Chl1, and Ctf4 (the TCCC pathway) (*Figure 7*) but independent of Mrc1 or Ctf18-RFC. These observations suggest that cohesin associated with un-replicated DNA is converted into cohesion through the agency of a very specific subset of proteins associated with the replisome. Further experiments will be required to establish whether this set includes other replication proteins previously implicated in cohesion such as Rmi1, Sgs1 and Top3 (*Lai et al., 2012*). It is not unlikely that essential replisome proteins are also

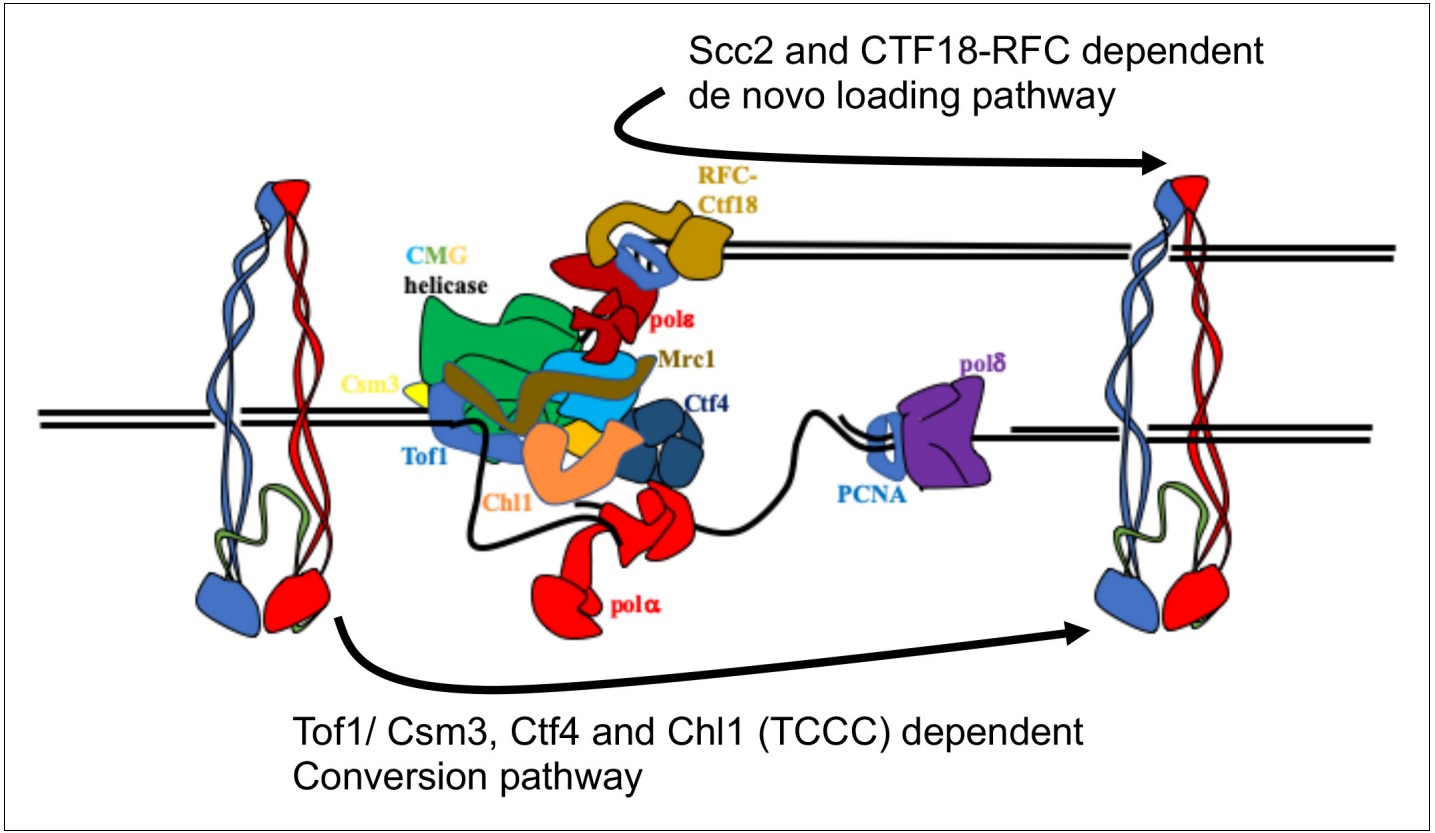

**Figure 7.** Two parallel pathways for cohesion establishment at the replication fork: chromosome associated cohesin is converted into cohesive structures during S phase. This process does not require Scc2 (except to load cohesin in the first place), but is strictly dependent on Tof1, Csm3, Chl1, and Ctf4 (the TCCC pathway). A second pathway that operates in parallel is concerned exclusively with de novo loading of cohesin onto nascent DNAs, in a Ctf18-RFC and Scc2-dependent manner (de novo pathway). See Discussion for details.

directly involved but addressing this question will require the generation of specific alleles that are defective in cohesion but not replication itself.

At present, the assay used to measure the conversion pathway requires the loading of non-cleavable cohesin not during the G1 immediately preceding replication but during the previous G2/M period. The reason for this admittedly baroque scheme is that Scc2 is normally required to maintain cohesin on chromosomes in G1 as well as to load it in the first place (*Srinivasan et al., 2019*), a feature that would preclude investigating its role during conversion. It was the chance discovery that chromosomal cohesin that had been loaded during G2/M does not require Scc2 to remain on chromosomes during the subsequent G1 period (*Srinivasan et al., 2019*), that allowed us to circumvent this limitation and to develop a rigorous assay for conversion. It is however conceivable that 'G2/M phase loaded' cohesin behaves differently to normal cohesin when cells undergo replication and that our discovery that conversion depends on the TCCC pathway only applies to cohesin that has inherited some modification generated exclusively during G2. We suggest that this is unlikely because the TCCC pathway also contributes to the establishment of cohesion by cohesin synthesised during G1 and presumably does so by converting chromosomal cohesin into cohesive structures. Another potential limitation of our assay is that it merely assays the fate of cohesin associated with a small *CEN*-based mini-chromosome. However, our observation that the TCCC pathway is required not only to generate mini-chromsome CDs but also to maintain cohesin's association with the entire genome when cells undergo S phase suggests that conversion acts genome wide.

Despite completely abrogating the conversion pathway, TCCC mutations are not lethal, have little adverse effect on mini-chromosome CD formation, and cause only modest, albeit highly

significant, cohesion defects on real chromosomes. Not all cohesion is therefore generated by conversion. There must be some other process by which cohesion is established. Our results together with previous findings *Xu et al., 2007* demonstrate that cohesion established in TCCC mutant cells depends on Ctf18-RFC. This form of cohesion is established from cohesin expressed during G1 or S phase. Cohesin loaded in G1 could have two fates: it could be a substrate for the TCCC pathway or be removed by Wapl-mediated releasing activity, neither of which are consistent with processing by the Ctf18 pathway. We therefore suggest that the latter is concerned exclusively with de novo loading of cohesin onto nascent DNAs (*Figure 7*), in which case there will be only a very narrow window of opportunity, a scenario consistent with the unexpected discovery that modest over-expression of Scc2/4 augments the Ctf18-RFC but not the TCCC pathway.

Our findings shed little insight into the molecular mechanisms of the two different pathways. In the case of conversion, CDs could in principle be created merely by passage of the replication fork through CMs that pre-exist ahead of the fork. Whether there is sufficient room for such passage is unclear. If not, then CDs must be created from CMs or from other forms of chromosomal cohesin rings that transiently open up to permit co-entrapment of sister DNAs. In the case of the de novo pathway, co-entrapment might involve entrapment of single stranded DNA associated with lagging strands by rings previously associated with double stranded leading strand DNA, a scenario consistent with the finding that Ctf18-RFC associates with the leading strand Pol ε polymerase (*Grabarczyk et al., 2018*). An Scc2-dependent reaction of this nature has recently been observed in vitro (*Murayama et al., 2018*). Because Eco1 is essential for Smc3 acetylation and thereby for maintaining all cohesion and because Ctf18-RFC or TCCC mutations merely reduce Smc3 acetylation (*Borges et al., 2013*), cohesion established by both pathways must be accompanied by Smc3 acetylation or in the case of vertebrate cells by Sororin binding. The nature of this linkage remains mysterious.

Because Ctf18-RFC and TCCC mutants have similarly severe cohesion defects both in mitosis (*Xu et al., 2007*) and meiosis (*Petronczki et al., 2004*), it appears that the de novo and conversion pathways make rather equal contributions to the establishment of cohesion. If true, this is either a remarkable coincidence or is instead a feature determined by properties intrinsic to the mechanism of DNA replication. Because Ctf18-RFC interacts with Pol ε while Ctf4 with Polα primase, it is tempting to speculate that the two reactions involve different intermediates. For example, during conversion cohesin might first be transferred to the lagging strand and only subsequently entrap the leading strand while during the de novo process cohesin might first load onto leading strand and only subsequently entrap the lagging strand, or vice versa. The equal contributions might therefore arise because there are equal opportunities to create the two different intermediates and they do not normally compete with each other. Ctf18-RFC and TCCC mutations also cause cohesion defects in mammalian cells (*Cortone et al., 2018*; *Zheng et al., 2018*), suggesting that both conversion and de novo pathways are conserved features of DNA replication in eukaryotic cells.

There are obvious parallels between the TCCC conversion pathway described here and the process by which histones associated with un-replicated DNAs are transferred to nascent ones (*Petryk et al., 2018*; *Yu et al., 2018*). Both types of process involve replisome components, some of which are shared (*Gan et al., 2018*). However, there is a fundamental difference, H3/H4 tetramers are transferred either to the leading or to the lagging strand whereas the cohesin conversion pathway connects leading and lagging strands. Whether cohesin is initially transferred asymmetrically to one strand before entrapping the second, as speculated above, remains to be established.

The fact that Ctf18-RFC and TCCC components are associated with replisomes does not exclude the possibility that either one or other pathway could in principle also act after replication has been completed. The fact that cohesin loaded during G2/M cannot form cohesion implies that this normally does not occur (*Haering et al., 2004*; *Srinivasan et al., 2019*; *Uhlmann and Nasmyth, 1998*). However, it has been reported that cohesion can be established genome wide in cells that have sustained DNA damage arising from a single unrepaired double strand break (*Ström and Sjögren, 2005*; *Unal et al., 2007*). It will be interesting to know whether this G2 cohesion pathway involves Ctf18-RFC, TCCC, or both. If in fact neither are required, then there must exist a third pathway involving yet another mechanism to generate cohesion.

# Materials and methods

**Key resources table**

| Reagent type (species) or resource | Designation | Source or reference | Identifiers | Additional information |
|---|---|---|---|---|
| Genetic reagent (*S. cerevisiae*) | NCBITaxon:4932 | This paper | Yeast strains | *Supplementary file 1* |
| Antibody | Mouse monoclonal Anti-V5 | BioRad | Cat# MCA1360 | (1:1000) |
| Antibody | Mouse monoclonal Anti PGK1 | ThermoFisher Scientific | Cat#459250 | (1:5000) |
| Chemical compound | Acid-washed glass beads | Sigma | Cat# G8722 | N/A |
| Chemical compound | ATP α-$^{32}$P | Hartmann Analytic | Cat# SRP-203 | N/A |
| Chemical compound | Bismaleimidoethane (BMOE) | ThermoFisher | Cat# 22323 | (5 mM) |
| Chemical compound | Complete EDTA free protease inhibitor cocktail | Roche | Cat# 4693132001 | (1:50 ml) |
| Chemical compound | Dithiothreitol | Fluka | Cat# BP172 | (5 mM) |
| Chemical compound | DMSO | Sigma | Cat# D8418 | N/A |
| Chemical compound | Immobilon Western ECL | Millipore | Cat# WBLKS0500 | N/A |
| Chemical compound | Trisodium citrate | Sigma | Cat# W302600 | N/A |
| Chemical compound | RNase A | Roche | Cat# 10109169001 | N/A |
| Chemical compound | Nocodazole | Sigma | Cat# M1404 | N/A |
| Chemical compound | PMSF | Sigma | Cat# 329-98-6 | N/A |
| Chemical compound | Potassium chloride | Sigma | Cat# P5405 | N/A |
| Chemical compound | Proteinase K | Roche | Cat# 03115836001 | N/A |
| Chemical compound | Sodium sulfite | Sigma | Cat# 71988 | N/A |
| Biological Sample | α-factor peptide | CRUK Peptide Synthesis Service | N/A | N/A |
| Commercial Assay or Kit | ChIP Clean and Concentrator Kit | Zymo Research | Cat# D5205 | N/A |
| Commercial Assay or Kit | E-Gel SizeSelect II Agarose Gels, 2% | ThermoFisher | Cat# G661012 | N/A |
| Commercial Assay or Kit | Library Quantification Kit Ion Torrent Platforms | KAPA Biosystems | Cat# 28-9537-67 | N/A |
| Commercial Assay or Kit | NEBNext Fast DNA library prep set for Ion Torrent | NEB | Cat# Z648094 | N/A |
| Commercial Assay or Kit | NuPAGE 3–8% Tris-Acetate Protein Gels, 1.5 mm, 10-well | ThermoFisher | Cat# E6270L | N/A |
| Commercial Assay or Kit | Prime-it II Random Primer Labelling Kit | Agilent | Cat# NP0321BOX | N/A |
| Commercial Assay or Kit | Protein G dynabeads | ThermoFisher | Cat# 300385 | N/A |
| Software, algorithm | Galaxy platform | *Giardine et al., 2005* | | |
| Software, algorithm | FastQC | Galaxy tool version 1.0.0 | https://usegalaxy.org | N/A |
| Software, algorithm | Trim sequences | Galaxy tool version 1.0.0 | https://usegalaxy.org | N/A |
| Software, algorithm | Filter FASTQ | Galaxy tool version 1.0.0 | https://usegalaxy.org | N/A |
| Software, algorithm | Bowtie2 | *Langmead and Salzberg, 2012* Galaxy tool version 0.2 | https://usegalaxy.org | N/A |

*Continued on next page*

*Continued*

| Reagent type (species) or resource | Designation | Source or reference | Identifiers | Additional information |
|---|---|---|---|---|
| Software, algorithm | Bam to BigWig | Galaxy tool version 0.1.0 | https://usegalaxy.org | N/A |
| Software, algorithm | Samtools | *Li et al., 2009* | https://usegalaxy.org | N/A |
| Software, algorithm | IGB browser | *Nicol et al., 2009* | http://samtools.sourceforge.net/ | N/A |
| Software, algorithm | Filter SAM or BAM | *Li et al., 2009* Galaxy tool version 1.1.0 | http://bioviz.org/igb/ | N/A |
| Software, algorithm | chr_position.py | This study | https://usegalaxy.org | N/A |
| Software, algorithm | filter.py | This study; *Petela, 2019* | https://github.com/naomipetela/nasmythlab-ngs | N/A |
| Software, algorithm | bcftools call | *Li et al., 2009* | https://github.com/naomipetela/nasmythlab-ngs | N/A |

## Contact for reagent and resource sharing

Further information and requests for resources and reagents should be directed to Madhusudhan Srinivasan (madhusudhan.srinivasan@bioch.ox.ac.uk) or Kim Nasmyth (ashley.nasmyth@bioch.ox.ac.uk).

## Method details

### Yeast cell culture

All strains are derivatives of W303 (K699). Strain numbers and relevant genotypes of the strains used are listed in the Key Resource Table. Cells were cultured at 25°C in YEP medium with 2% glucose unless stated otherwise.

To arrest the cells in G1, α-factor was added to a final concentration of 2 mg/L, every 30 min for 2.5 hr. Cells were released from G1 arrest by filtration wherein cells were captured on 1.2 μm membrane (Whatman GE Healthcare), washed with 1 L YEPD and resuspended in the appropriate fresh media.

To arrest cells in G2, nocodazole (Sigma) was added to the growth media to a final concentration of 10 μg/mL and cells were incubated until synchronization was achieved (>95% large-budded cells).

### Growth conditions for cohesin conversion assay

The cells expressing non cleavable Scc1 under the control of GAL promoter were grown in SC-TRP (Raffinose) medium at 25°C to $OD_{600nm}$ = approximately 0.5 and arrested in G2 phase. Galactose was added to the culture to a final concentration of 2% and the cultures were incubated for 45 min at 25°C. The cultures were released from the G2 arrest by filtration, wherein the cells were captured on 1.2 μm membrane (Whatman GE Healthcare), washed with 1 L YEPD and resuspended to a density of $OD_{600nm}$ = 0.2 in fresh YEPD media containing α-factor (2 mg/L) and incubated at 25°C. 30 min later the cultures were filtered and resuspended in YEPD media containing α-factor (2 mg/L) and incubated at 25°C for 60–90 min (>95% small unbudded cells). Cells were released from G1 arrest into the subsequent G2 phase by filtration and resuspension in YEPD medium containing nocodazole.

To inactivate Scc2 and Ctf8 during S phase (using *scc2-45* and *ctf8-17* temperature sensitive alleles), the cultures grown in SC-TRP Raffinose medium were arrested in G1 at 25°C. 30 min prior to release from the G1 arrest galactose was added (2% final). The cultures were released into YEP Raffinose media containing nocodazole and galactose(2% final) that was pre-warmed to 37°C, followed by incubation at 37°C.

## In vivo chemical crosslinking (For minichromosome IP)

40 OD units of the cells were washed in ice-cold PBS and re-suspended in 1 mL ice-cold PBS. The suspensions were treated with 42 µL BMOE (stock: 125 mM in DMSO, 5 mM final) and incubated for 6 min on ice. Cells were washed with 2 × 2 mL ice-cold PBS containing 5 mM DTT and the pellets were flash frozen in liquid nitrogen and stored at −80°C.

## Minichromosome IP

The frozen pellets were resuspended in 800 µL lysis buffer (25 mM Hepes pH 7.5, 100 mM KCl, 1 mM MgCl$_2$, 10 mM trisodium citrate, 25 mM sodium sulfite, 0.25% triton-X, freshly supplemented with Roche Complete Protease Inhibitors (2X) and PMSF (1 mM), lysed in a FastPrep-24 (MP Biomedicals) for 3 × 1 min at 6.5 m/s with 800 µl of acid-washed glass beads (425–600 µm, Sigma) and lysates cleared (10 min, 13.2 k*g*). Protein concentrations were normalized after Bradford assay and cohesin immuno-precipitated by incubation with anti-PK antibody (BioRad, 2 hr, 4°C) and protein G dynabeads (2 hr, 4°C, with rotation). After cohesin immuno-precipitation, protein G dynabeads were washed with 1 × 1 ml lysis buffer, 1 × 1 ml wash buffer (10 mM Tris-HCl, pH 8.0; 0.25 M LiCl; 0.5 % NP-40; 0.5% sodium deoxycholate; 1 mM EDTA; 1 mM PMSF) and 1x with TE pH7.5. Beads were resuspended in 30 µl 1% SDS with DNA loading dye, incubated at 65°C for 4 min and the supernatant run on a 0.8% agarose gel containing ethidium bromide (1.4 V/cm, 22 hr, 4°C). After Southern blotting using alkaline transfer, bands were detected using a 32 P labeled TRP1 probe. All the minichromosome IPs presented in this manuscript are representative of at least two independent biological replicates.

## SDS gel electrophoresis and western blotting

Whole cell lysates were resolved in NuPAGE3–8% or 4–12% gradient gels (ThermoFisher Scientific) and transferred onto PVDF membranes using the Trans-blot Turbo transfer system (BioRad). For visualization, the membrane was incubated with Immobilon Western Chemiluminescent HRP substrate (Millipore) before detection using an ODYSSEY Fc Imaging System (LI-COR).

## Sister chromatid cohesion assay

Cells grown non synchronously in YEPD medium at 25°C were fixed with 4% formaldehyde for 45 min at room temperature, washed 2X with PBS sorbitol (1xPBS, 1M Sorbitol) and stored at 4°C. In situ immunofluorescence to detect MYC tagged Securin and GFP marked *URA3* loci was carried out as described in *Silver, 2009*. Briefly, the fixed cells were spheroplasted by treatment with Zymolyase (100T) for 30 min at 30°C. The spehroplasts were adhered onto a polylysine coated slide and permiabilised by incubation with 1% NP-40 for 5 min. The slides were blocked with PBS containing 1% BSA and incubated overnight with PBS/1% BSA/anti MYC antibody at 4°C. The slides were washed 10X with PBS/1% BSA and treated with fluorescently labelled secondary antibody for 2 hr at room temperature. The slides were mounted with DAPI containing mounting medium and observed with a Zeiss Axio Imager.Z1 microscope (63 × objective, NA = 1.40) equipped with a coolSNAP HQ camera.

## Fitness assays

Relative fitness is a measure of the reproductive success of one genotype compared to a reference genotype, measured over the course of several generations. When genetic manipulations affect chromosome metabolism, the outcome of competition two populations of microbial cells which are competing for the resources present in batch cultures, is mostly influenced by the doubling time during exponential phase. To measure relative fitness, strains were competed against *ctf4Δ*, *ctf18Δ* (*Figure 6B*), or wild type (*Figure 6—figure supplement 1A*) reference strains. A *pFA6a* plasmid expressing yCerulean (reference strains) or mCherry (test strains) fluorescent proteins under the *ACT1* promoter was digested with *Age*I and integrated in *ACT1* loci. For measuring the relative fitness, 10 ml of YPD were inoculated in individual glass tubes with either the frozen reference or test strains. After 24 hr. the strains were mixed in fresh 10 ml YPD tubes at a ratio dependent on the expected fitness of the test strain compared to the reference (i.e. 1:1 if believed to be nearly equally fit) and allowed to proliferate at 30°C for 24 hr. 10 µl of samples were taken from this mixed culture (day 0) and the ratio of the two starting strains was immediately measured. Mixed cultures were

then propagated by diluting them 1:1000 into fresh medium every 24 hr for 4 days, monitoring the strain ratio at every passage. Strain ratios and number of generations occurred between samples were measured by flow cytometer (Fortessa, BD Bioscience, RRID:SCR_013311, Franklin Lakes, NJ). Ratios *r* were calculated based on the number of fluorescent and non-fluorescent yCerulean events detected by the flow cytometer:

$$r = \frac{NonFluorescent_{events}}{Fluorescent_{events}}$$

Generations between time points *g* were calculated based on total events measured at time 0 hr. and time 24 hr.:

$$g = \frac{\log_{10}(events_{t24}/events_{t0})}{\log_{10}2}$$

Linear regression was performed between the $(g, \log_e r)$ points relative to every sample. Relative fitness was calculated as the slope of the resulting line. To account for potential fitness differences due to fluorescent protein expression, or unintended mutations, the relative fitness between isogenic strains carrying different fluorophores was subtracted from other comparisons involving any of them. The mean relative fitness *s* was calculated from measurements obtained from at least three independent biological replicates. Error bars represent standard deviations. The p-values reported in figures are the result of *t*-tests assuming unequal variances (Welch's test). Note that the absolute values of relative fitness change depending on the reference strain used.

## FACS analysis

Approximately $0.5 \times 10^7$ cells were sedimented at 13 K g for 30 s, and pellets were fixed with 1 ml 50% ethanol and stored at 5°C. The fixed cells were spun at 6 k g and the pellets resuspended in 1 ml 50 mM Tris-HCl (pH 7.5) + 20 µl of 10 mg/ml RNaseA and incubated with shaking at 37°C overnight. Cells were pelleted and resuspended in 500 µl PI buffer (200 mM Tris-HCl [pH 7.5], 211 mM NaCl, 78 mM MgCl2) and propidium iodide was added at 50 µg/ml final concentration. Samples were sonicated for 5 s at 40% power and 50–100 µl was diluted into 1 ml 50 mM Tris-HCl (pH 7.5) and read with a Becton Dickinson FACSCalibur, ensuring 30,000 events per sample.

## Calibrated ChIP-sequencing

Cells were grown exponentially to OD$_{600x00A0}$ = 0.5 and the required cell cycle stage where necessary. 15 OD$_{600nm}$ units of *S. cerevisiae* cells were then mixed with 5 OD$_{600nm}$ units of *C. glabrata* to a total volume of 45 mL and fixed with 4 mL of fixative (50 mM Tris-HCl, pH 8.0; 100 mM NaCl; 0.5 mM EGTA; 1 mM EDTA; 30% (v/v) formaldehyde) for 30 min at room temperature (RT) with rotation.

The fixative was quenched with 2 mL of 2.5 M glycine (RT, 5 min with rotation). The cells were then harvested by centrifugation at 3,500 rpm for 3 min and washed with ice-cold PBS. The cells were then resuspended in 300 µL of ChIP lysis buffer (50 mM Hepes-KOH, pH 8.0; 140 mM NaCl; 1 mM EDTA; 1% (v/v) Triton X-100; 0.1% (w/v) sodium deoxycholate; 1 mM PMSF; 2X Complete protease inhibitor cocktail (Roche)) and an equal amount of acid-washed glass beads (425–600 µm, Sigma) added before cells were lysed using a FastPrep−24 benchtop homogeniser (MP Biomedicals) at 4°C (3 × 60 s at 6.5 m/s or until >90% of the cells were lysed as confirmed by microscopy).

The soluble fraction was isolated by centrifugation at 2,000 rpm for 3 min then sonicated using a bioruptor (Diagenode) for 30 min in bursts of 30 s on/30 s off at high level in a 4°C water bath to produce sheared chromatin with a size range of 200–1000 bp. After sonication the samples were centrifuged at 13,200 rpm at 4°C for 20 min and the supernatant was transferred into 700 µL of ChIP lysis buffer. 30 µL of protein G Dynabeads (Invitrogen) were added and the samples were pre-cleared for 1 hr at 4°C. 80 µL of the supernatant was removed (termed 'whole cell extract [WCE] sample') and 5 µg of antibody (anti-PK (Bio-Rad) or anti-HA (Roche)) was added to the remaining supernatant which was then incubated overnight at 4°C. 50 µL of protein G Dynabeads were then added and incubated at 4°C for 2 hr before washing 2x with ChIP lysis buffer, 3x with high salt ChIP lysis buffer (50 mM Hepes-KOH, pH 8.0; 500 mM NaCl; 1 mM EDTA; 1% (v/v) Triton X-100; 0.1% (w/v) sodium deoxycholate; 1 mM PMSF), 2x with ChIP wash buffer (10 mM Tris-HCl, pH 8.0; 0.25 M LiCl; 0.5 % NP-40; 0.5% sodium deoxycholate; 1 mM EDTA; 1 mM PMSF) and 1x with TE pH7.5. The

immunoprecipitated chromatin was then eluted by incubation in 120 μL TES buffer (50 mM Tris-HCl, pH 8.0; 10 mM EDTA; 1% SDS) for 15 min at 65°C and the collected supernatant termed 'IP sample'. The WCE samples were mixed with 40 μL of TES3 buffer (50 mM Tris-HCl, pH 8.0; 10 mM EDTA; 3% SDS) and all samples were de-crosslinked by incubation at 65°C overnight. RNA was degraded by incubation with 2 μL RNase A (10 mg/mL; Roche) for 1 hr at 37°C and protein was removed by incubation with 10 μL of proteinase K (18 mg/mL; Roche) for 2 hr at 65°C. DNA was purified using ChIP DNA Clean and Concentrator kit (Zymo Research).

## Preparation of sequencing libraries

Sequencing libraries were prepared using NEBNext Fast DNA Library Prep Set for Ion Torrent Kit (New England Biolabs) according to the manufacturer's instructions. Briefly, 10–100 ng of fragmented DNA was converted to blunt ends by end repair before ligation of the Ion Xpress Barcode Adaptors. Fragments of 300 bp were then selected using E-Gel SizeSelect2% agarose gels (Life Technologies) and amplified with 6–8 PCR cycles. The DNA concentration was then determined by qPCR using Ion Torrent DNA standards (Kapa Biosystems) as a reference. 12–16 libraries with different barcodes could then be pooled together to a final concentration of 350 pM and loaded onto the Ion PI V3 Chip (Life Technologies) using the Ion Chef (Life Technologies). Sequencing was then completed on the Ion Torrent Proton (Life Technologies), typically producing 6–10 million reads per library with an average read length of 190 bp.

## Data analysis, alignment and production of BigWigs

Unless otherwise specified, data analysis was performed on the Galaxy platform. Quality of reads was assessed using FastQC (Galaxy tool version 1.0.0) and trimmed as required using 'trim sequences' (Galaxy tool version 1.0.0). Generally, this involved removing the first 10 bases and any bases after the 200th, but trimming more or fewer bases may be required to ensure the removal of kmers and that the per-base sequence content is equal across the reads. Reads shorter than 50 bp were removed using Filter FASTQ (Galaxy tool version 1.0.0, minimum size: 50, maximum size: 0, minimum quality: 0, maximum quality: 0, maximum number of bases allowed outside of quality range: 0, paired end data: false) and the remaining reads aligned to the necessary genome(s) using Bowtie2 (Galaxy tool version 0.2) with the default (–sensitive) parameters (mate paired: single-end, write unaligned reads to separate file: true, reference genome: SacCer3 or CanGla, specify read group: false, parameter settings: full parameter list, type of alignment: end to end, preset option: sensitive, disallow gaps within *n*-positions of read: 4, trim *n*-bases from 5' of each read: 0, number of reads to be aligned: 0, strand directions: both, log mapping time: false).

To generate alignments of reads that uniquely align to the *S. cerevisiae* genome, the reads were first aligned to the *C. glabrata* (CBS138, genolevures) genome with the unaligned reads saved as a separate file. These reads that could not be aligned to the *C. glabrata* genome were then aligned to the *S. cerevisiae* (sacCer3, SGD) genome and the resulting BAM file converted to BigWig (Galaxy tool version 0.1.0) for visualization. Similarly, this process was done with the order of genomes reversed to produce alignments of reads that uniquely align to *C. glabrata*.

## Visualisation of ChIP-seq profiles

The resulting BigWigs were visualized using the IGB browser. To normalize the data to show quantitative ChIP signal the track was multiplied by the samples' occupancy ratio (OR) and normalized to 1 million reads using the graph multiply function. In order to calculate the average occupancy at each base pair up to 60 kb around all 16 centromeres, the BAM file that contains reads uniquely aligning to *S. cerevisiae* was separated into files for each chromosome using 'Filter SAM or BAM' (Galaxy tool version 1.1.0). A pileup of each chromosome was then obtained using samtools mpileup (Galaxy tool version 0.0.1) (source for reference list: locally cached, reference genome: SacCer3, genotype likelihood computation: false, advanced options: basic). These files were then amended using our own script (chr_position.py) to assign all unrepresented genome positions a value of 0. Each pileup was then filtered using another in-house script (filter.py) to obtain the number of reads at each base pair within up to 60 kb intervals either side of the centromeric CDEIII elements of each chromosome. The number of reads covering each site as one successively moves away from these CDEIII elements

could then be averaged across all 16 chromosomes and calibrated by multiplying by the samples' OR and normalizing to 1 million reads.

## Data and software availability

### Scripts

All scripts written for this analysis method are available to download from https://github.com/naomi-petela/nasmythlab-ngs (*Petela, 2019*; copy archived at https://github.com/elifesciences-publications/nasmythlab-ngs).

*Chr_position.py* takes mpileups for *S. cerevisiae* chromosomes and fills in gaps, with each position in the chromosome added given a read depth of 0.

*Filter60.py* reads the files produced by Chr_position.py and takes the read depth for all positions 60 kb either side of the CDEIII for all chromosomes, produces an average for each position and multiples it by the OR. The OR should be derived from the reads aligned in the appropriate bam files (*Hu et al., 2015*).

### Calibrated ChIP-seq data

The calibrated ChIP-seq data (raw and analyzed data) have been deposited on GEO under accession number GSE151551.

## Additional information

### Funding

| Funder | Grant reference number | Author |
|---|---|---|
| Wellcome | 107935/Z/15/Z | Kim A Nasmyth |
| Cancer Research UK | 26747 | Kim A Nasmyth |

The funders had no role in study design, data collection and interpretation, or the decision to submit the work for publication.

### Author contributions

Madhusudhan Srinivasan, Conceptualization, Data curation, Formal analysis, Validation, Investigation, Visualization, Methodology, Writing - original draft, Project administration, Writing - review and editing; Marco Fumasoni, Resources, Data curation, Formal analysis, Validation, Investigation, Visualization, Methodology, Writing - original draft; Naomi J Petela, Data curation, Software, Formal analysis, Validation, Visualization; Andrew Murray, Supervision, Funding acquisition, Project administration, Writing - review and editing; Kim A Nasmyth, Conceptualization, Funding acquisition, Writing - original draft, Project administration, Writing - review and editing

### Author ORCIDs

Madhusudhan Srinivasan (iD) https://orcid.org/0000-0001-5676-4219
Marco Fumasoni (iD) http://orcid.org/0000-0002-4507-7824
Kim A Nasmyth (iD) https://orcid.org/0000-0001-7030-4403

### Decision letter and Author response

Decision letter https://doi.org/10.7554/eLife.56611.sa1
Author response https://doi.org/10.7554/eLife.56611.sa2

## Additional files

### Supplementary files

- Source data 1. Source data for the Relative fitness assay.
- Supplementary file 1. Strain list.
- Transparent reporting form

## Data availability

Sequencing data have been deposited in GEO under accession number GSE151551.

The following dataset was generated:

| Author(s) | Year | Dataset title | Dataset URL | Database and Identifier |
|---|---|---|---|---|
| Srinivasan M, Fumasoni M, Petela NJ, Murray A, Nasmyth KA | 2020 | Cohesion is established during DNA replication utilising chromosome associated cohesin rings as well as those loaded de novo onto nascent DNAs | https://www.ncbi.nlm.nih.gov/geo/query/acc.cgi?acc=GSE151551 | NCBI Gene Expression Omnibus, GSE151551 |

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
