## [Decision Letter]

**Acceptance summary:**

This paper elegantly demonstrates that cohesion establishment in yeast occurs through two redundant pathways: de novo cohesin loading and cohesin recycling. This is a major step forward in understanding how the cohesion establishment reaction, which is essential for chromosome segregation, occurs.

**Decision letter after peer review:**

Thank you for submitting your article "Sister chromatid cohesion is established at the replication fork by two independent pathways operating in parallel" for consideration by *eLife*. Your article has been reviewed by three peer reviewers, and the evaluation has been overseen by a Reviewing Editor and Kevin Struhl as the Senior Editor. The following individuals involved in review of your submission have agreed to reveal their identity: Jean-Paul Javerzat (Reviewer #2); Ana Losada (Reviewer #3).

The reviewers have discussed the reviews with one another and the Reviewing Editor has drafted this decision to help you prepare a revised submission. In recognition of the fact that revisions may take longer than the two months we typically allow, until the research enterprise restarts in full, we will give authors as much time as they need to submit revised manuscripts.

This is a foundational study for our understanding of how sister chromatid cohesion is established during DNA replication. The data lead to a model that two pathways coexist in yeast cells: one that requires de novo loading of cohesin by Scc2 in the wake of the replication fork and a second, Scc2-independent, pathway that converts cohesin complexes loaded onto unreplicated DNA ahead of the fork into cohesive structures behind it. This work addresses a long-standing question that has been very difficult to address and a clear strength of this work is the use of many elegant genetic tricks.

Summary:

Sister chromatid cohesion is thought to be generated at the replisome during S-phase. Meanwhile, cohesin binding to chromosomes is dependent on the Scc2/Scc4 loading complex, which is active before, during, and after DNA replication. This manuscript addresses the longstanding and important question of whether cohesion is built from G1-loaded cohesin, requires de novo loaded cohesin at the replisome, or both. A previous study suggested two independent and partially redundant pathways for generating cohesion. Here the authors show convincing evidence that both pathways generate sister-chromatid cohesion: the TCCC pathway converts G1-loaded cohesin into cohesive cohesin, while Ctf18-RFC is required for Scc2-dependent cohesion establishment reaction within the replisome. To do this, the authors use an assay to monitor cohesion establishment in minichromosomes and an elegant experimental design that allows them to load tagged cohesin complexes in G2 that remain on chromatin until the next round of DNA replication.

Essential revisions:

1) All experiments were performed in cells in which Wapl-dependent release has been abrogated. A previous study showed that inactivating Wapl results in the permanent activation of the DNA Damage Induced (DI) cohesion pathway (Mol Cell. 2009 May 15;34(3):311-21). This raises the possibility that the phenomena described here include some aspect of the DI-pathway, i.e., the authors actually observed a mixture of S-phase and DI pathways for generating cohesive cohesin. The authors should discuss this point. Related to this, in some experiments it would be important to remind readers that what is being monitored is the behavior of the separase-insensitive, Wapl-insensitive cohesin loaded in previous G2. For instance: "half or more chromosomal cohesin disappeared from all parts of genome when *chl1Δ* cells underwent S phase (Figure 4B)". One possibility is to use a superscript to refer to this cohesin, same as a different color (red) is used for it in the figures. Did the authors check by ChIP whether the presence of "regular" cohesin (loaded in G1) is affected after passage of the replication fork in *chl1Δ* mutants?

2) The experiments presented in Figure 6 address the importance of the two described pathways in cohesion. That these two pathways existed was proposed before, and cohesion had been tested in mutants for the different components of both pathways as well as combinations. What is new and striking is that one of the pathways does not depend on Scc2. In a previous publication by Fumasoni and Murray, these authors show the effect of adding extra copies of Scc2 to *ctf4Δ* mutants in cohesion and fitness. Part of the experiment (fitness) is repeated here, presumably testing in parallel the fitness of *ctf4Δ* and *ctf18Δ* mutants. It is expected that extra copies of Scc2 improve cohesion in *ctf4Δ* but not in *ctf18Δ*. Why was cohesion not tested in this experiment as well?

3) The relative fitness experiments are methodologically sound. However, this approach is still uncommon in the field (despite its advantages over comparative methods). Therefore, the concept of fitness and how the experiment is done should be briefly described in main text.

The authors should also provide the raw data for these experiments either in a supplement or on Mendeley. This will help other to follow this approach as well as check the calculations.

4) The turn-over of G2-loaded, NC-6C cohesin. Figure 2 uses 6C non cleavable cohesin loaded during the previous cell cycle (2C SMC1 2C SMC3 and GALp-2C SCC1NC induced in G2/M) to show that Scc2 is not required for CDs generation in the next S-phase. The authors used calibrated ChIP sequencing to check that cohesin remained bound during the G1 arrest (Figure 2—figure supplement 1). The data are shown as a ratio (T=0 / T=60min). Therefore, we expect to see a ratio of 100% if no changes occurred. However, this is not what is observed on the graph. The median is around 100%, consistent with cohesin remaining bound, but there are great variations along the chromosome, suggesting cohesin sliding or, unloading and reloading cycles. Although it does not invalidate the main conclusion that Scc2 is dispensable for CDs generation, it casts doubt about the absence of 6C cohesin turnover. The authors should comment on this.

Also, in previous publications, the authors have supplemented this ratio with the actual ChIP enrichment of each condition across the chromosome (Srinivasan et al., 2019), please provide this in the current manuscript.

5) The importance of DNA replication for the observed effects. The authors propose that cohesion can be established by de novo cohesin loading at the replisome, but as presented, the data do not provide direct evidence that DNA replication is required. A cohesin / BrdU ChIP-seq (or several loci analysed by q-PCR) in the experiment scheme shown Figure 5C may provide evidence that cohesin binding to replicated domain is Ctf8-dependent. Similarly, based on the data shown in Figure 4, it is suggested that cohesin might be released by the replisome when the conversion pathway is not functional. The experiment shows that cohesin is not DNA-bound after S-phase but does not provide evidence that release is due to replication. A Cdc6-degron experiment might tell whether DNA replication per se is required. To argue that cohesin release takes place at the replisome, a cohesin / BrdU ChIP-seq (or several loci analysed by q-PCR) from S-phase cells may provide evidence that cohesin is still bound to unreplicated DNA but reduced/absent from replicated domains. While such experiments might be out of the scope of the current manuscript, the authors should ensure that their conclusions take this point into consideration throughout the manuscript and they should also consider changing the title.

6) In Figure 4—figure supplement 1C and Figure 5—figure supplement 1B, the authors use a Southern blot for TRP1 sequence to show that the plasmid is present in equal amounts in each of the genetic conditions. They only show a set of four bands cropped in each case. Each of these lanes should have at least two types of DNA species. There should be signal from the endogenous trp1-1 locus and signal from the plasmid. Please show the full lane with markers to differentiate between the different DNAs recognized by the probe.

---

## [Author Response]

Essential revisions:1) All experiments were performed in cells in which Wapl-dependent release has been abrogated. A previous study showed that inactivating Wapl results in the permanent activation of the DNA Damage Induced (DI) cohesion pathway (Mol Cell. 2009 May 15;34(3):311-21). This raises the possibility that the phenomena described here include some aspect of the DI-pathway, i.e., the authors actually observed a mixture of S-phase and DI pathways for generating cohesive cohesin. The authors should discuss this point.

Regarding permanent activation of DI cohesion pathway in wapl mutants: We can certainly discuss this point. However, we would like to point out that there is no evidence that loss of Wapl actually activates a DI cohesion pathway. It is more likely that the DI pathway inactivates Wapl-mediated release which would otherwise remove cohesin that had formed cohesion during G2. Crucially, we show that inactivation of Wapl-mediated release does not in fact permit cohesion establishment as measured by CD formation. In other words, we do not detect significant CD formation when 6C cohesin is expressed in G2 arrested cells defective in Wapl-mediated release. The whole issue may therefore be irrelevant to our conclusions.

Related to this, in some experiments it would be important to remind readers that what is being monitored is the behavior of the separase-insensitive, Wapl-insensitive cohesin loaded in previous G2. For instance: "half or more chromosomal cohesin disappeared from all parts of genome when chl1D cells underwent S phase (Figure 4B)". One possibility is to use a superscript to refer to this cohesin, same as a different color (red) is used for it in the figures. Did the authors check by ChIP whether the presence of "regular" cohesin (loaded in G1) is affected after passage of the replication fork in chl1D mutants?

We thank the reviewer for this suggestion, we can include a reminder that we are following the fate of the non-cleavable cohesin loaded in the previous G2. We cannot however check the fate of “regular” cohesin expressed in G1 because Separase remains active in G1 in yeast cells. Previous studies have found that steady state level of wild type cohesin on chromosomes is reduced by approximately 50% in *chl1∆* mutants when compared to *CHL1* (Samora et al., 2016). Modified the main text for clarity, this pool of cohesin is referred to as Scc1^NC^ cohesin (expressed in the G2 phase of the previous cell cycle).

2) The experiments presented in Figure 6 address the importance of the two described pathways in cohesion. That these two pathways existed was proposed before, and cohesion had been tested in mutants for the different components of both pathways as well as combinations. What is new and striking is that one of the pathways does not depend on Scc2. In a previous publication by Fumasoni and Murray, these authors show the effect of adding extra copies of Scc2 to ctf4∆ mutants in cohesion and fitness. Part of the experiment (fitness) is repeated here, presumably testing in parallel the fitness of ctf4∆ and ctf18∆ mutants. It is expected that extra copies of Scc2 improve cohesion in ctf4∆ but not in ctf18∆. Why was cohesion not tested in this experiment as well?

Fair point. We were in the process of building the strains to test the prediction that increased fitness of a *ctf18∆* mutant resulting from extra copies of SCC2 co-relates with a reduction in cohesion defects of the *ctf18∆* mutant.

3) The relative fitness experiments are methodologically sound. However, this approach is still uncommon in the field (despite its advantages over comparative methods). Therefore, the concept of fitness and how the experiment is done should be briefly described in main text.The authors should also provide the raw data for these experiments either in a supplement or on Mendeley. This will help other to follow this approach as well as check the calculations.

Fair comment. We can certainly do this. The relative fitness experiment is now briefly described in the main text and raw data is presented as a Source data 1.

4) The turn-over of G2-loaded, NC-6C cohesin. Figure 2 uses 6C non cleavable cohesin loaded during the previous cell cycle (2C SMC1 2C SMC3 and GALp-2C SCC1NC induced in G2/M) to show that Scc2 is not required for CDs generation in the next S-phase. The authors used calibrated ChIP sequencing to check that cohesin remained bound during the G1 arrest (Figure 2—figure supplement 1). The data are shown as a ratio (T=0 / T=60min). Therefore, we expect to see a ratio of 100% if no changes occurred. However, this is not what is observed on the graph. The median is around 100%, consistent with cohesin remaining bound, but there are great variations along the chromosome, suggesting cohesin sliding or, unloading and reloading cycles. Although it does not invalidate the main conclusion that Scc2 is dispensable for CDs generation, it casts doubt about the absence of 6C cohesin turnover. The authors should comment on this.

Certainly, the variation in cohesin levels along the chromosome could for instance be a result of cohesin re-localisation. Commented on this in the main text

Also, in previous publications, the authors have supplemented this ratio with the actual ChIP enrichment of each condition across the chromosome (Srinivasan et al., 2019), please provide this in the current manuscript.

Certainly, this will be provided as figure supplements. Actual ChIP profiles now presented in and Figure 4—figure supplement 2.

5) The importance of DNA replication for the observed effects. The authors propose that cohesion can be established by de novo cohesin loading at the replisome, but as presented, the data do not provide direct evidence that DNA replication is required. A cohesin / BrdU ChIP-seq (or several loci analysed by q-PCR) in the experiment scheme shown Figure 5C may provide evidence that cohesin binding to replicated domain is Ctf8-dependent.

It is important to remember that even in the absence of both conversion and de novo pathways, Scc2 dependent cohesin loading onto individual DNAs is not affected at all (Figure 5C). It will be impossible to distinguish cohesin that is loaded onto individual DNAs (that happens throughout the cell cycle, including S phase) from cohesive cohesin that holds sister DNAs. The prediction will be that in a cohesin / BrdU ChIP-seq experiment one should detect cohesin on nascent DNA irrespective of the presence of active de novo pathway due to Scc2 dependent loading onto individual nascent DNAs. We have however considered permanently modifying chromosome associated cohesin (biotinlylation) and following the fate of the modified cohesin upon DNA replication in order to study the conversion pathway (PNAS October 8, 2019 116 (41) 20605-20611). Such experiments however are beyond the scope of the current study.

Similarly, based on the data shown in Figure 4, it is suggested that cohesin might be released by the replisome when the conversion pathway is not functional. The experiment shows that cohesin is not DNA-bound after S-phase but does not provide evidence that release is due to replication. A Cdc6-degron experiment might tell whether DNA replication per se is required.

Fair point, we had considered this issue and had already constructed *chl1∆* and *CHL1* strains where Cdc45 can be depleted to inhibit DNA replication in order to check if the reduction in cohesin levels we observe in *chl1∆* mutants depends on DNA replication per se. However, this is complex and sophisticated experiment that cannot be performed in our lab in the near future. Crucially, our central conclusions stand independent of the result of such an experiment.

To argue that cohesin release takes place at the replisome, a cohesin / BrdU ChIP-seq (or several loci analysed by q-PCR) from S-phase cells may provide evidence that cohesin is still bound to unreplicated DNA but reduced/absent from replicated domains. While such experiments might be out of the scope of the current manuscript, the authors should ensure that their conclusions take this point into consideration throughout the manuscript and they should also consider changing the title.

Our title reads: ‘Cohesion is established during DNA replication by converting pre-existing chromosomal cohesin into cohesive structures as well as by de novo loading of cohesin onto nascent DNAs’. We believe we have provided evidence to justify the title and do not believe that we have overinterpreted our data. We do not claim a direct involvement of the replisome in cohesin eviction in the absence of the conversion pathway, we have been careful to state only that this happens as cells undergo S phase.

6) In Figure 4—figure supplement 1C and Figure 5—figure supplement 1B, the authors use a Southern blot for TRP1 sequence to show that the plasmid is present in equal amounts in each of the genetic conditions. They only show a set of four bands cropped in each case. Each of these lanes should have at least two types of DNA species. There should be signal from the endogenous trp1-1 locus and signal from the plasmid. Please show the full lane with markers to differentiate between the different DNAs recognized by the probe.

Under the conditions we extract the genomic DNA and electrophorese the gels, vast majority of the genomic DNA remains in or very close to the wells. We routinely cut the top part of the gels before southern blotting. Author response image 1 is an example of an uncropped membrane from a southern blot of strains harbouring the TRP mini-chromosome (Note: the wells were removed from this example as well). The top bands correspond to fragments of the genomic trp1-1 locus while the bottom band is the supercoiled mini-chromosome DNA. We have shown the cropped southern blot of the genomic DNA simply to make the point that under conditions cohesin fails to IP the mini-chromosome DNA, the cells stably maintain the mini-chromosomes during the course of the experiment. We are not sure what showing the full membrane would add. Moreover, we have shown that this effect we observed in the mini-chromosome IP is true for the entire genome by Calibrated-ChIP Sequencing.
